# Mapping global leaf inclination angle (LIA) based on field measurement data

Sijia Li[1,2,3], Hongliang Fang[2,3]

[1]National-Local Joint Engineering Laboratory of Geo-Spatial Information Technology, Hunan University of Science and Technology, Xiangtan 411201, China

[2]LREIS, Institute of Geographic Sciences and Natural Resources Research, Chinese Academy of Sciences, Beijing 100101, China

[3]College of Resources and Environment, University of Chinese Academy of Sciences, Beijing 100049, China

*Correspondence to*: Sijia Li (lisj.19b@igsnrr.ac.cn)

**Abstract.** Leaf inclination angle (LIA), the angle between leaf surface normal and zenith directions, is a vital trait in radiative transfer, rainfall interception, evapotranspiration, photosynthesis, and hydrological processes. Due to the difficulty in obtaining large-scale field measurement data, LIA is typically assumed to follow the spherical leaf distribution or simply considered constant for different plant types. However, the appropriateness of these simplifications and the global LIA distribution are still unknown. This study compiled global LIA measurements and generated the first global 500 m mean LIA (MLA) product by gap-filling the LIA measurement data using a random forest regressor. Different generation strategies were employed for noncrops and crops. The MLA product was evaluated by validating the nadir leaf projection function (G(0)) derived from the MLA product with high-resolution reference data. The global MLA is 41.47°±9.55°, and the value increases with latitude. The MLAs for different vegetation types follow the order of cereal crops (54.65°) > broadleaf crops (52.35°) > deciduous needleleaf forest (50.05°) > shrubland (49.23°) > evergreen needleleaf forest (47.13°) ≈ grassland (47.12°) > deciduous broadleaf forest (41.23°) > evergreen broadleaf forest (34.40°). Cross-validation shows that the predicted MLA presents a medium consistency (*r* = 0.75, RMSE = 7.15°) with the validation samples for noncrops, whereas crops show relatively lower correspondence (*r* = 0.48 and 0.60 for broadleaf crops and cereal crops) because of limited LIA measurements and strong seasonality. The global G(0) distribution is opposite to that of the MLA and agrees moderately with the reference data (*r* = 0.62, RMSE = 0.15). This study shows that the common spherical and constant LIA assumptions may underestimate the intercept capability for most vegetation. The MLA and G(0) products derived in this study would enhance our knowledge about global LIA and should greatly facilitate remote sensing retrieval and land surface modeling studies.

The global MLA and G(0) products can be accessed at:

Li, S. and Fang, H. 2024, https://doi.org/10.5281/zenodo.12739662.

## 1 Introduction

Vegetation regulates terrestrial carbon and water cycles through a series of biophysical processes such as photosynthesis, respiration, and transpiration (Foley et al., 1996; Chen et al., 2019). These biophysical processes are mainly carried by leaves and the characterization of leaves within canopies is vital for remote sensing and earth system modeling (Ross, 1975; Lawrence et al., 2019). Leaf inclination angle (LIA) denotes the inclination of the leaf or needle to the horizontal plane or the angle between the leaf surface normal and zenith (Wilson, 1960). LIA is a key canopy structural trait that determines radiative transfer, rainfall interception, evapotranspiration, photosynthesis, and hydrological processes (Sellers, 1985; Ross, 1981; Mantilla-Perez and Salas Fernandez, 2017; Xiao et al., 2000; Maes and Steppe, 2012). LIA has been used in radiative transfer modeling (RTM), remote sensing inversion, and land surface modeling (LSM) studies (Tang et al., 2016; Wang and Fang, 2020; Lawrence et al., 2019; Ross, 1975).

At the canopy scale, the probability density of LIA or the fraction of leaf area per unit LIA is expressed as the leaf angle distribution (LAD) (De Wit, 1965). De Wit (1965) summarized six theoretical LADs, including planophile, erectophile, extremophile, plagiophile, uniform, and spherical distributions. Specifically, the spherical distribution assumes that the relative probability density of the LIA is proportional to the area of the corresponding sphere surface element and its mean leaf inclination angle (MLA) equals 57.3° (MLA = 57.3°) (De Wit, 1965). Furthermore, Ross (1981) defined the inclination index ($\chi_L$) to describe the departure of LAD from the spherical distribution. For the planophile distribution, $\chi_L = 1$; for the erectophile distribution, $\chi_L = -1$; and for the spherical distribution, $\chi_L = 0$. In the radiative transfer regime, LIA is generally represented by the leaf projection function ($G(\theta)$), which is defined as the average projection ratio of unit leaf area in the illumination or viewing direction $\theta$ (Ross, 1981; Nilson, 1971). The spherical distribution is characterized by an isotropic leaf projection function ($G \equiv 0.5$) (De Wit, 1965).

In the field, LIA can be measured directly based on the leaf's geometrical structure or using indirect optical methods (Lang, 1973; Ryu et al., 2010; Norman and Campbell, 1989; Weiss and Baret, 2017). Using these methods, several LIA measurements have been carried out and some LIA datasets were constructed (Kattge et al., 2020; Chianucci et al., 2018; Hinojo-Hinojo and Goulden, 2020; Pisek and Adamson, 2020). These field methods are usually time-consuming and labor-intensive and are typically difficult to acquire large-scale LIA (Li et al., 2023). In addition, the existing LIA datasets have not been comprehensively analyzed. LIA has also been estimated from satellite imagery through empirical relationships or radiative transfer model inversions (Zou and Mõttus, 2015; Bayat et al., 2018; Goel and Thompson, 1984). Remote sensing methods are used primarily for crops in local regions, and the generality of these algorithms is limited (Li et al., 2023). Due to the difficulty in large-scale LIA measurements and estimations, our knowledge about the global LIA remains lacking.

Because our understanding of the global LIA is limited, different LIA simplification strategies have been adopted in various studies. For example, LIA is typically assumed to follow the spherical distribution (Tang et al., 2016; Zhao et al., 2020; Wang and Fang, 2020). However, this assumption may decrease the accuracy of radiative transfer modeling, significantly underestimate the radiation interception (Stadt and Lieffers, 2000), and cause large errors (>50%) in leaf area index (LAI)

measurements and inversions (Yan et al., 2021). The spherical LIA assumption may introduce greater error in the nadir direction than other viewing geometries (Yan et al., 2021), considering the large G variation in this direction (Wilson, 1959). The lack of global LIA knowledge also limits the retrieval of other vegetation structural parameters(Li et al., 2023). In many LSMs, LIA is commonly treated as a fixed value for different plant function types (PFT) (Lawrence et al., 2019; Majasalmi and Bright, 2019). Field LIA measurements have demonstrated that the spherical distribution is not appropriate for forests, and the PFT-dependent LIA ignores LIA variation within the PFT (Pisek et al., 2013; Yan et al., 2021; Majasalmi and Bright, 2019).

This study aims to generate the first global MLA map from existing LIA field measurements using a data-driven gap-filling method. This method involves spatial expansion and upscaling of LIA measurements, and a random forest regressor using input spectral, climate, and PFT data. Based on the global MLA map, we tested whether the spherical LIA assumption is appropriate at the global scale. The new MLA map was validated by comparing the nadir G (G(0)) derived from the MLA with high-resolution reference data. Section 2 outlines the materials and methods employed to generate and evaluate the global MLA. Section 3 presents the global LIA measurements, global MLA and G(0), and evaluation results. Section 4 discusses the performance of the global MLA and G(0), the usage of the new MLA map, and the limitations of the study. Section 5 presents the main conclusions.

## 2 Materials and methods

### 2.1 LIA measurement data

#### 2.1.1 TRY LIA dataset

TRY is a network of vegetation scientists headed by Future Earth, the Max Planck Institute for Biogeochemistry, and German Centre for Integrative Biodiversity Research, providing a global database of curated plant traits (the TRY database) (https://www.trydb.org/TryWeb/Home.php). Since its establishment in 2007, the TRY database has continuously evolved and has become one of the most widely used vegetation trait databases. The latest V6 version (released on October 13, 2022) employed in this study contains 15,409,681 trait records covering 305,594 plant taxa (Kattge et al., 2020). In this database, LIA was recorded as a numerical or categorical variable. After data extraction and checking, 31,043 valid records were used, which include numerical LIA, locations, and species. Many measurements lack location information, whereas, for some locations, there are many measurements for individual species. The spatial distribution map appears relatively sparse despite a large volume of data (Fig. 1). The LIA measurements in South America are mainly from palms while the LIA measurements of most species are located in the Northern Hemisphere.

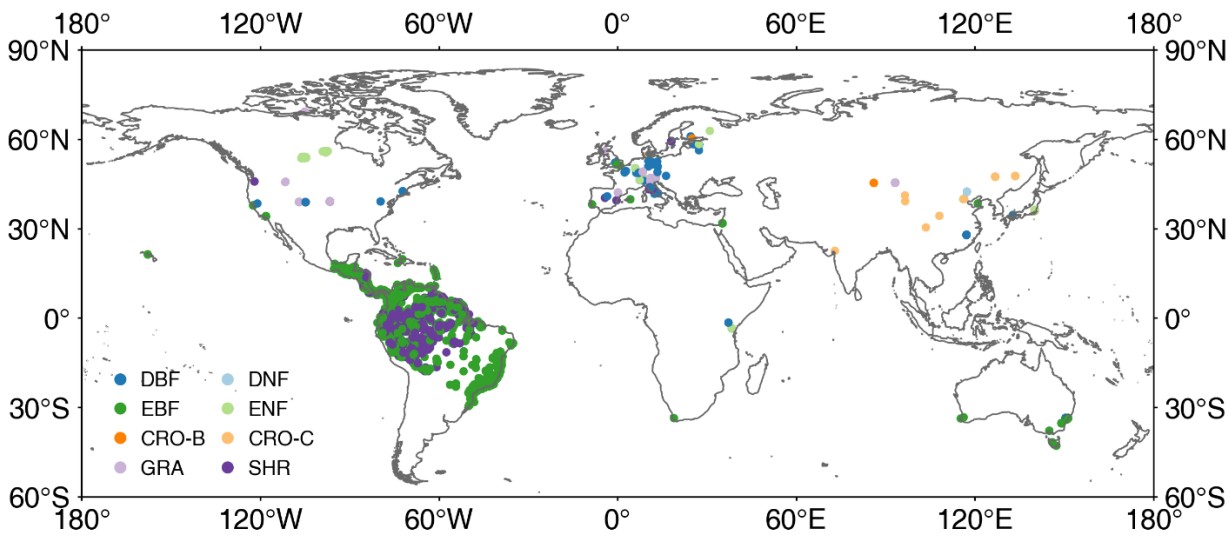

**Figure 1.** The locations of global leaf inclination angle measurements. DBF: deciduous broadleaf forest, DNF: deciduous needleleaf forest, EBF: evergreen broadleaf forest, ENF: evergreen needleleaf forest, CRO-B: broadleaf crops, CRO-C: cereal crops, GRA: grassland, SHR: shrubland.

### 2.1.2 LIA data from the literature

The LIA measurements in published literature were collected via keyword search (leaf angle, leaf inclination angle, and leaf tilt angle) in the Web of Science, Google Scholar, Google, and Chinese documentary databases. The LIA, location, and species information were manually extracted from the literature (Fig. 1). Several LIA measurements were already included in the TRY database (Chianucci et al., 2018; Pisek and Adamson, 2020). After aggregating LIA measurements for the same species at the same location, 780 LIA records were accessed from previous studies (Hinojo-Hinojo and Goulden, 2020; Pisek et al., 2022; Chen et al., 2021).

### 2.1.3 Manual LIA extraction

Only a few measurements in the northern tundra region were obtained, and the measurements in tropical regions are dominated by palm trees (Fig. 1). Therefore, LIA data for the northern tundra and tropical regions were extracted from horizontal side-view photographs searched from Google (Fig. S1). ImageJ software (https://imagej.nih.gov/ij/) was used to process the leveled photographs and derive LIA following the method of Pisek et al. (2011). The TRY species location data (848,919, Fig. S3b) (Jan 03, 2022) were used to obtain the dominant species information in tropical rainforests and the northern tundra. The species location points in these two vegetation types were spatially filtered and the frequency of occurrence for each species was counted. The species with a high frequency of occurrence were selected to measure the LIA. For each species, more than 75 leaves perpendicular to the viewing direction were selected and processed based on visual

judgment to ensure the stability and reliability of the MLA (Pisek et al., 2013). In total, the MLA of 104 species was
manually derived.
In this study, most LIA measurements are obtained with protractor and level digital photogrammetry, especially for
needleleaf species. Therefore, the distinction between branches and leaves is considered. The diverse LIA records from
different sources were sorted to match the TRY species and to get the PFT based on the TRY Categorical Traits Dataset
2018 (https://www.try-db.org/TryWeb/Data.php#3). LIA measurements from different sources were unified into canopy-
level MLA with average operation by leaf number (see Appendix A). If there were multiple LIA records for the same species,
the mean value was computed for the same location and species. In total, 5,554 LIA records of 1,194 species were collected,
covering the growing season from 2001 to 2022. LIA location replicates per species range from 1 to 330, and most replicates
(98 %) are less than 50. Considering the different numbers of records for each species, the LIA data was further aggregated
by species.
**2.2 Remote sensing data**
**2.2.1 Ancillary data used for MLA mapping**
The ancillary data used for global MLA mapping and analysis are listed in Table 1. Most earth observation data were
accessed and processed in Google Earth Engine (GEE) (https://earthengine.google.com/). The PFT classification system in
the MODIS global 500 m land cover type product (MCD12Q1 C6) was used and mode-aggregated from 2001 to 2022 to
match the LIA measurements (Fig. S2) (Sulla-Menashe et al., 2019). The 2001–2022 Landsat surface reflectance (Level 2,
Collection 2, Tier 1) (Crawford et al., 2023), including Landsat 5 (2001–2012), Landsat 7 (2012–2013), and Landsat 8
(2013–2022) was utilized to generate a global 30 m PFT map (Section 2.3.1), which was subsequently employed for LIA
upscaling. Considering the sensitivity of directional reflectance variation to LIA (Jacquemoud et al., 2009; Li et al., 2023),
the 2001–2022 MODIS bidirectional reflectance distribution function (BRDF) model parameters dataset (MCD43A1 C6.1)
(Schaaf and Wang, 2015b) and nadir BRDF adjusted reflectance dataset (MCD43A4 V6 NBAR) (Schaaf and Wang, 2015a)
produced daily using 16 days of Terra and Aqua MODIS data at 500 m resolution and were utilized as predictive variables.
We used MCD43A1 C6.1 and MCD12Q1 and MCD43A4 C6 for MLA mapping as these data were available on GEE when
this study was conducted. Only minor calibration changes and polarization correction were adopted in the upgrading from
Collection    6    to    6.1,    while    the    MCD12Q1    and    MCD43A4    algorithms    remain    the    same
(https://landweb.modaps.eosdis.nasa.gov/data/userguide/MODIS_Land_C61_Changes.pdf).    In    addition,    the    multi-year
aggregation of these products (Table 2) further mitigates the version impact. Due to the scarcity of crop LIAs and the lack of
location information for existing crop LIA measurements, fine-resolution (10/30 m) crop-type maps (Table 1) in 2018 were
employed to support crop LIA mapping. Other data include the ERA5-Land reanalysis data, the ALOS digital elevation
model (AW3D30 V3.2), and the 2001–2022 MODIS LAI product (MCD15A2H) (Myneni, 2015). The LAI product was
averaged and aggregated from 2001–2022.

**Table 1.** Remote sensing data for global MLA mapping. BRDF: bidirectional reflectance distribution function.

| Category | Data | Year | Spatial resolution | Temporal resolution | Reference |
|---|---|---|---|---|---|
| Plant function type | MCD12Q1 C6 | 2001–2022 | 500 m | Yearly | (Sulla-Menashe et al., 2019) |
| Surface reflectance | Landsat collection 2 | 2001–2022 | 30 m | 16 days | (Crawford et al., 2023) |
| | MCD43A4 V6 NBAR | 2001–2022 | 500 m | Daily | (Schaaf and Wang, 2015a) |
| BRDF | MCD43A1 C6.1 | 2001–2022 | 500 m | Daily | (Schaaf and Wang, 2015b) |
| Crop type | Cropland Data Layers (CDL) | 2018 | 30 m | Yearly | (Boryan et al., 2011) |
| | EUCROPMAP | 2018 | 10 m | Yearly | (D'andrimont et al., 2021) |
| | AAFC Annual Crop Inventory | 2018 | 30 m | Yearly | (Fisette et al., 2013) |
| | Northeast China crop-type map | 2018 | 30 m | Yearly | (You et al., 2021) |
| | NESEA-Rice10 | 2018 | 10 m | Yearly | (Han et al., 2021) |
| | China maize map | 2018 | 30 m | Yearly | (Shen et al., 2022) |
| | China winter wheat map | 2018 | 30 m | Yearly | (Dong et al., 2020) |
| Climate | ERA5-Land | 2001–2022 | 0.1° | Monthly | (Muñoz-Sabater et al., 2021) |
| Terrain | AW3D30 V3.2 | — | 30 m | — | (Tadono et al., 2014) |

**2.2.2 High-resolution reference data**
The high-resolution reference datasets provided by Ground Based Observations for Validation (GBOV,
https://land.copernicus.eu/global/gbov/dataaccessLP/) and DIRECT 2.1 (https://calvalportal.ceos.org/lpv-direct-v2.1) were
used to evaluate the generated global MLA (Fig. 2). These datasets provide high-resolution (20/30 m) LAI, effective LAI
(LAIe), and fractional vegetation cover (FVC) data over a 3 km × 3 km area centered on each site generated using empirical
relationships between various vegetation indices and ground measurements (Li et al., 2022; Brown et al., 2020). GBOV has
provided continuous high-resolution reference data since 2013 (Fig. 2).

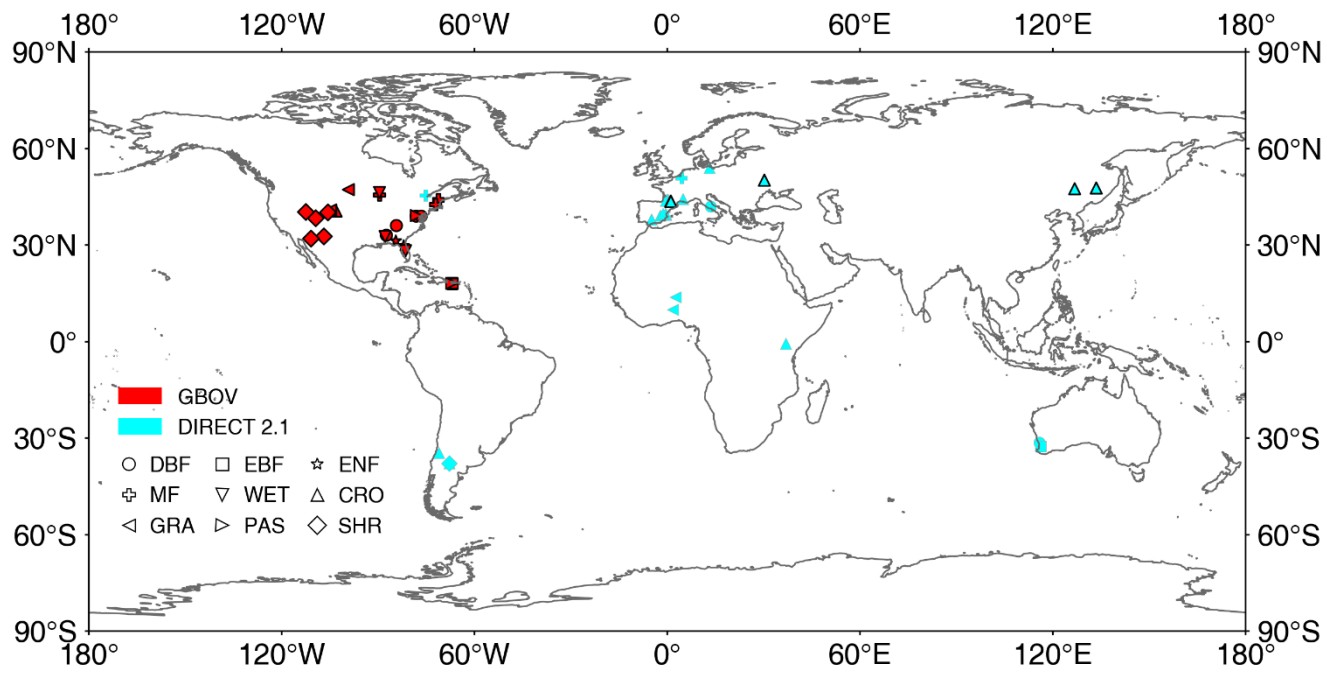


**Figure 2.** Locations of GBOV and DIRECT 2.1 sites used in this study. CRO: Cultivated crops, MF: Mixed forest, PAS: Pasture/hay,
WET: Woody wetlands. See Fig. 1 for other acronyms. The black frame indicates those sites with >5 continuous records.
**2.3 Mapping global LIA**
**2.3.1 Data preparation**
Many studies have treated LIA as a species-specific static trait and ignored within-species variations when LIA
measurements are limited (Pisek et al., 2022; Toda et al., 2022; Raabe et al., 2015). Following the rationale, the spatial
coverage of LIA measurements was expanded, and those records without location information were utilized (section 2.1.1).
Under this assumption, the LIA measurements were expanded through TRY species location data with species name
matching. The species location data comprises trait measurements for common species representing a hundreds-of-square-
meter area around the location. The dominant species was artificially identified by investigators and thus the spatial
representativeness is considered. When a species had multiple LIA observations at different locations, the nearest LIA was
assigned to the TRY species location. Visual inspections were conducted to remove potential TRY location biases, especially
for non-vegetated points such as water bodies and deserts. After spatial expansion, the number of LIAs reached 12,328 and
its spatial distribution became more uniform (Fig. S3c).
In this study, the scale gap between field measurements and satellite remote sensing data was fully considered. The canopy
level MLA measurement is regarded as equal to 30 m-MLA considering its spatial representativeness. To upscale the MLA
measurements from canopy level to the satellite resolution (500 m), a 30 m PFT map was first derived from Landsat
reflectance using a random forest classification method. The random forest was trained at a 500 m scale using the mode-
aggregated MODIS PFT classification map as training samples to generate a 30 m PFT map by hierarchically selecting
homogeneous pixels (with a coefficient of variation in reflectance < 0.2). The classification features were the same as those
in the MODIS classification algorithm (Sulla-Menashe et al., 2019). For a 500 m pixel with multiple PFTs (Fig. 3a), when
one PFT had no MLA measurement, the MLA of the PFT was assigned with the value of its nearest neighbor within 100 km
with the same PFT. This distance setting (100 km) was based on a previous study that derived global maps for various leaf
traits from a limited number of field measurements, remote sensing, and climate data (Moreno-Martínez et al., 2018). In field
measurement, the entire canopy MLA is commonly calculated as the average of all measured leaf LIAs weighted by leaf area
(see Appendix A) (Zou et al., 2014; De Wit, 1965; Yan et al., 2021). Leaves with larger areas have higher weights.
Upscaling MLA from 30 m to 500 m follows the same rationale as that from leaf to canopy scale. For a 30 m pixel with a
higher LAI, the weight of the pixel is higher. Therefore, the 500 m MLA was computed as the weighted average of the
enhanced vegetation index (EVI2) assuming a linear relationship between LAI and EVI2 (see Appendix A) (Dong et al.,
2019; Alexandridis et al., 2019). Although previous studies have reported that vegetation index may be nonlinearly
correlated to LAI because of the saturation effect at medium and high LAI conditions, EVI2 is highly resistant to the
saturation effect (Gao et al., 2023). The errors caused by this slight nonlinearity were further analyzed in Section 4.4.
$$MLA_{500m} = \frac{\sum MLA_{30m} \times EVI2_{30m}}{\sum EVI2_{30m}} \hspace{4cm} (1)$$

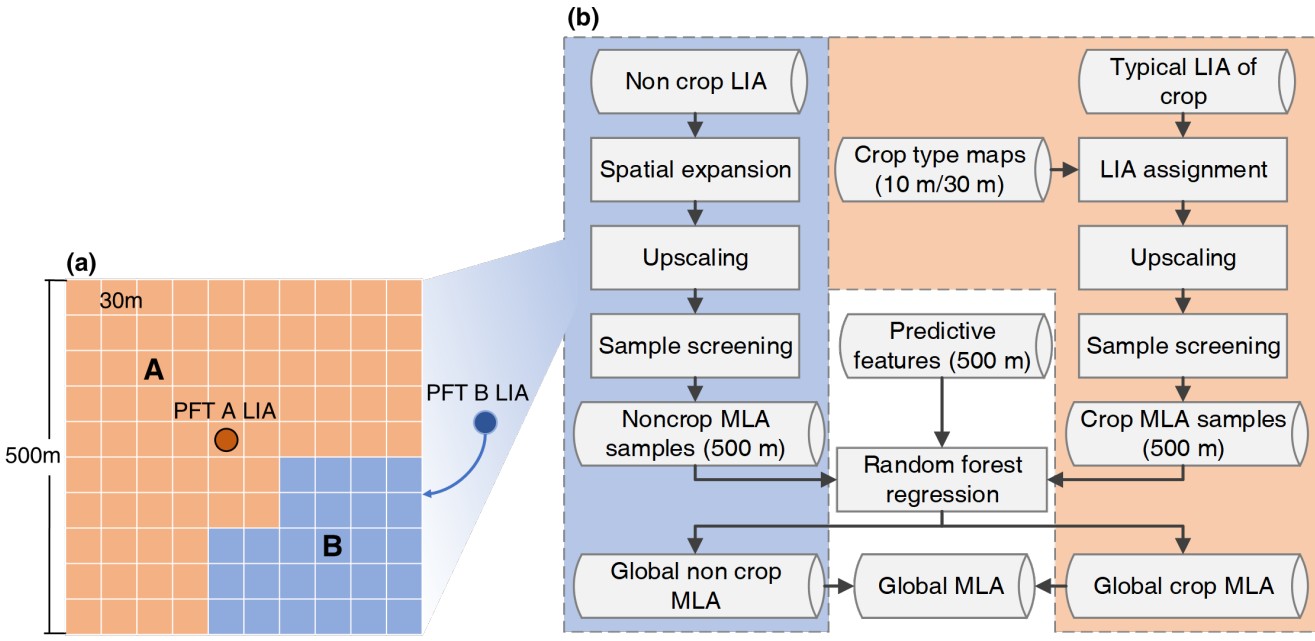

**Figure 3.** Leaf inclination angle (LIA) upscaling (a) and global mean LIA (MLA) mapping (b) strategies.
The 500 m upscaled MLA samples were further refined to select the most representative samples following three criteria: 1)
the coefficient of variation of the 30 m EVI2 in the 500 m pixel is less than 0.2, 2) the vegetation proportion in the 500 m
pixel is greater than 0.8, and 3) the proportion of PFTs represented by the MLA measurements in the 500 m pixel is greater
than 0.4. The final number of samples after refinement is 3,013 with a uniform spatial distribution (Fig. 4).

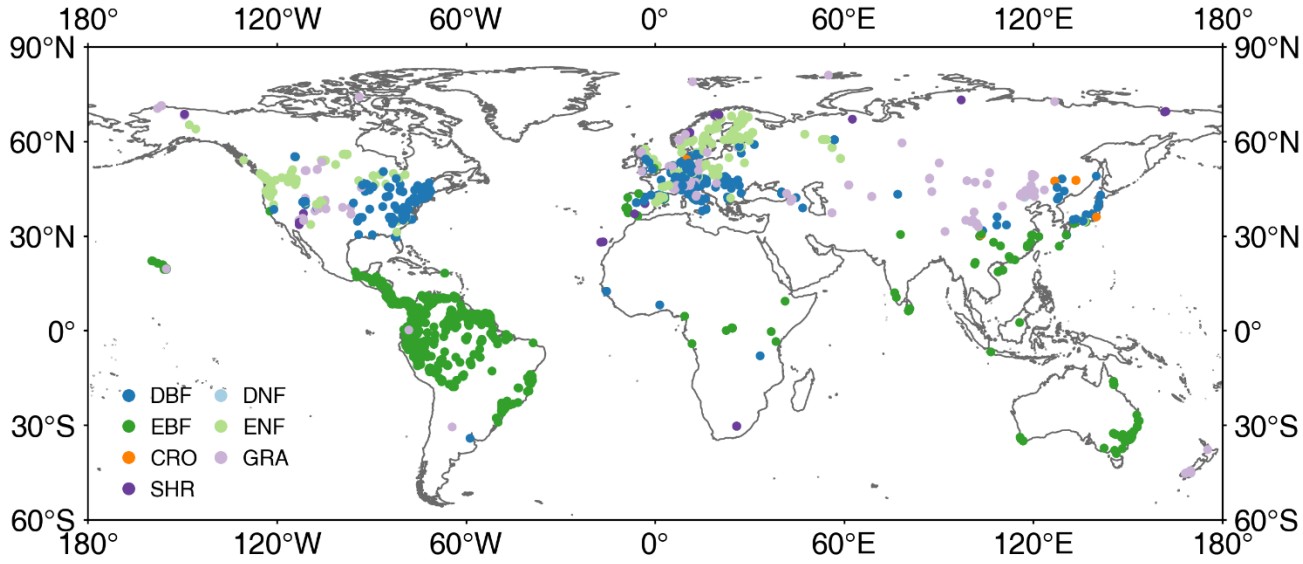

**Figure 4.** Distribution of global mean leaf inclination angle samples after screening. See Fig. 1 for acronyms.
**2.3.2 Global MLA mapping**
Different mapping strategies were employed for noncrops and crops (Fig. 3b) considering the small number of valid crop
samples (Fig. 4) and the lack of location information for most crop samples. For noncrops, the upscaled 500 m MLA
samples were used to train a random forest regressor to predict the global MLA from different features (Table 2). All input
features were unified to the 500 m resolution. Therefore, the derived MLA map corresponds to the average MLA at the 500
m scale. Notably, this study used all MODIS BRDF and spectral reflectance data including low-quality ones that may be
contaminated by clouds. The multi-year aggregation (Table 2) can partly mitigate the influence induced by low-quality
observations (Sulla-Menashe et al., 2019). Normalized difference vegetation index (NDVI) was used as the predictive
feature because it is strongly coupled with LIA, especially under low and medium vegetation density conditions (Dong et al.,
2019; Zou and Mõttus, 2015). To reduce computational complexity and potential overfitting, a feature selection process was
conducted based on the variable importance (the sum of the decrease in Gini impurity index over all trees in the forest)
computed by the model, and only the 40 most important variables were used in the final prediction. During the training
process, the out-of-bag error was minimized to obtain the optimal hyperparameters. The prediction performance of the
random forest regressor was evaluated using a ten-fold cross-validation approach with upscaled MLA samples.
For crops, the measured MLA values were averaged for different crop types as a typical MLA (Table S2). After assigning
typical MLAs for different crops with high-resolution crop maps (Table 1), the high-resolution crop MLA were upscaled to
500 m as training samples (Eq. (1)). Only the samples with a crop area ratio > 80% within a 500 m pixel were selected for
training. The crops were further divided into broadleaf crops and cereal crops and processed with the same procedure used
for noncrops (Fig. 3b). All procedures were conducted on GEE under the WGS-84 geographic coordinate system.
Two quality layers were added to represent the quality of input data and the prediction model. The input data quality was
denoted by the proportion of high-quality BRDF inversions for each pixel. The prediction model quality was represented
qualitatively for each pixel considering whether the MLA was predicted by extrapolating beyond the range of the training
samples. The random forest model is typically regarded as a black-box and its uncertainty is difficult to quantify in the
present study.

**Table 2.** Predictive features in global MLA mapping.

| Category | Features | Variables | Number |
|----------|----------|-----------|--------|
| Spectral | Blue, green, red, near-infrared reflectance | 10%, 33%, 50%, 67%, 90% quantiles and standard deviation | 24 |
| | NDVI | 10%, 33%, 50%, 67%, 90% quantiles and standard deviation | 6 |
| BRDF | Kernel coefficients of the red band | 10%, 33%, 50%, 67%, 90% quantiles and standard deviation | 18 |
| | Kernel coefficients of near-infrared band | 10%, 33%, 50%, 67%, 90% quantiles and standard deviation | 18 |
| PFT | PFT | Constant | 1 |
| Climate | Solar downward radiation | Mean and standard deviation | 2 |
| | Temperature | Mean and standard deviation | 2 |
| | Precipitation | Mean and standard deviation | 2 |
| Terrain | Elevation | Constant | 1 |
| | Slope | Constant | 1 |
| | Aspect | Constant | 1 |

**2.4 Evaluation of global MLA**
The global MLA map was indirectly evaluated using the nadir leaf projection function, because of the lack of high-resolution
reference MLA. G(0) is important because it is coherent with the satellite nadir observations. The global G(0) was derived
from the MLA and evaluated with high-resolution reference following the upscaling scheme recommended by the Land
Product Validation (LPV) Subgroup of the Committee on Earth Observation Satellites (CEOS) (http://lpvs.gsfc.nasa.gov/).
Assuming a single-parameter ellipsoidal leaf angle distribution (Campbell, 1990; Wang et al., 2007), the parameter $\chi$, the
ratio of the horizontal and vertical axes of an ellipsoid, was first derived from MLA in radians. Compared to other models,
the single-parameter ellipsoidal leaf angle distribution is a relatively more accurate and simpler model and has been used in
many remote sensing studies (Campbell, 1990; Wang et al., 2007; Kuusk, 2001; Verhoef et al., 2007).
$\chi = -3 + (\frac{MLA}{9.65})^{-0.6061}$        (2)
The G(θ) value in the nadir direction (θ=0°) was calculated using the following analytical formula.
$G(\theta) = \frac{\sqrt{(\chi^2 + tan^2\,\theta)}\,cos\,\theta}{\chi + 1.774(\chi + 1.182)^{-0.73}}$        (3)
The reference G(0) was derived from high-resolution LAI, FVC, and clumping index (CI) (=LAIe/LAI) with the Beer-
Lambert law (Fig. S4) (Nilson, 1971).
$P(\theta) = exp^{-\frac{G(\theta)*LAI*CI(\theta)}{cos(\theta)}}$        (4)
Where $P(\theta)$, $CI(\theta)$, and $G(\theta)$ denote the gap fraction, CI, and G in direction $\theta$, respectively. Specifically, the gap fraction in
the nadir direction can be expressed by FVC.
$P(0) = 1 - FVC$        (5)
Therefore, the reference G(0) was derived using the following formula.
$G(0)\_CI(0) = -\frac{ln(1-FVC)}{CI(0)*LAI}$        (6)
By using the whole CI as the nadir CI (CI(0)) in the above equation (Fang et al., 2021; Li et al., 2022), G(0) was calculated
as follows:
$G(0)\_CI \approx -\frac{ln(1-FVC)}{CI*LAI}$        (7)
The MLA product was first upscaled to 3 km through a weighted averaging method using the MODIS LAI to derive G(0)
(Eq. (3)). The reference LAI, FVC, and CI were also upscaled to 3 km through simple averaging to compute the reference
G(0) (Eq. (7)). The MLA-derived G(0) and the reference G(0) were compared at the 3 km × 3 km area around each site. The
correlation coefficient ($r$), bias, and root mean square error (RMSE) were calculated as the evaluation metrics, as follows:
$r = \sqrt{1 - \frac{\sum_{i=1}^{n}(\hat{y}_i - y_i)^2}{\sum_{i=1}^{n}(y_i - \bar{y})^2}}$        (8)
$Bias = \frac{1}{n}\sum_{i=1}^{n}(\hat{y} - y_i)$        (9)
$RMSE = \sqrt{\frac{1}{n}\sum_{i=1}^{n}(\hat{y} - y_i)^2}$        (10)
where $\hat{y}_i$, $y_i$, and $n$ denote the MLA-derived G(0), reference G(0), and the number of G(0), respectively.

## 3 Results

### 3.1 Global measured LIA values

The species-aggregated LIA was employed in the analysis of global LIA measurements. Fig. 5 shows the distributions of global measured LIA values for different PFTs. The global measured MLA is 40.74° and generally follows the order of CRO-C > GRA > ENF > CRO-B > EBF > SHR > DNF > DBF (Table 3). Cereal crops exhibit the highest MLA (59.11°), whereas DBF has the most horizontal leaves (MLA = 34.94°). GRA and EBF show large LIA variations (Std = 20.44° and 17.17°), whereas CRO-B exhibits a small range. The DNF LIA measurements are only for one species and show very little variation (Fig. 5).

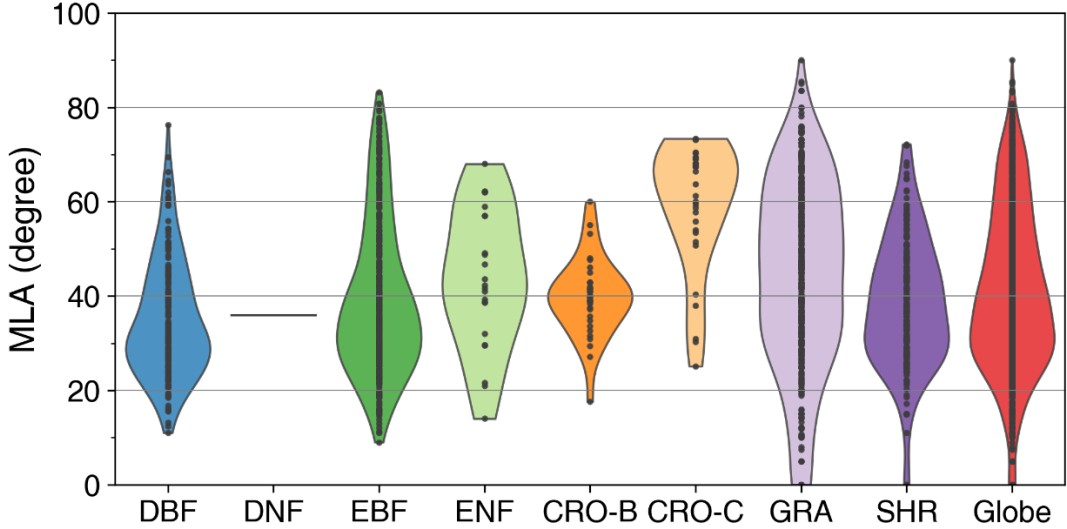

**Figure 5.** Distribution of global mean LIA (MLA) for different plant function types (see Fig. 1 for acronyms). The last shape shows the global average. Statistics are conducted for each species as represented by points in the figure.

**Table 3.** Statistics of leaf inclination angle measured for different plant functional types (PFT). STD is the standard deviation. The inclination index ($\chi_L$) is converted from mean leaf inclination angle (MLA) ($\chi_L = 2cos(\textbf{MLA}) - 1$) (Lawrence et al., 2019).

| PFT | DBF | DNF | EBF | ENF | CRO-B | CRO-C | GRA | SHR | Globe |
|---|---|---|---|---|---|---|---|---|---|
| Number of species | 171 | 1 | 347 | 23 | 32 | 31 | 399 | 190 | 1194 |
| Mean(°) | **34.94** | 35.88 | 39.30 | 43.69 | 39.71 | **59.11** | 44.13 | 38.32 | 40.74 |
| STD (°) | 12.40 | 0.00 | **16.11** | 14.40 | 8.11 | 13.28 | **20.17** | 13.80 | 17.12 |
| $\chi_L$ | 0.64 | 0.62 | 0.55 | 0.45 | 0.54 | 0.03 | 0.44 | 0.57 | 0.52 |

## 3.2 The relationships between MLA and other variables

Fig. 6 shows the importance of the top 40 variables in the MLA prediction obtained from the random forest regression model. The importance of these 40 variables accounts for 78% of the total importance among all 76 variables. Spectral features account for 30% of the importance, which is higher than that of other features. Among the spectral features, NDVI, near-infrared (NIR) band, and red band reflectance are most critical for MLA prediction. The importance of BRDF features is comparable to that of climatic variables (21% vs. 20%), followed by terrain features (7%). Among the BRDF features, the NIR BRDF information shows a higher contribution than the red band, with importance in the following order: geometrically scattered kernel> isotropic scattering kernel > volumetric scattering kernel. The importance ranking of the climatic variables follows the order of precipitation ≈ solar radiation > temperature. In addition, elevation, slope, and aspect significantly impact on the MLA prediction.

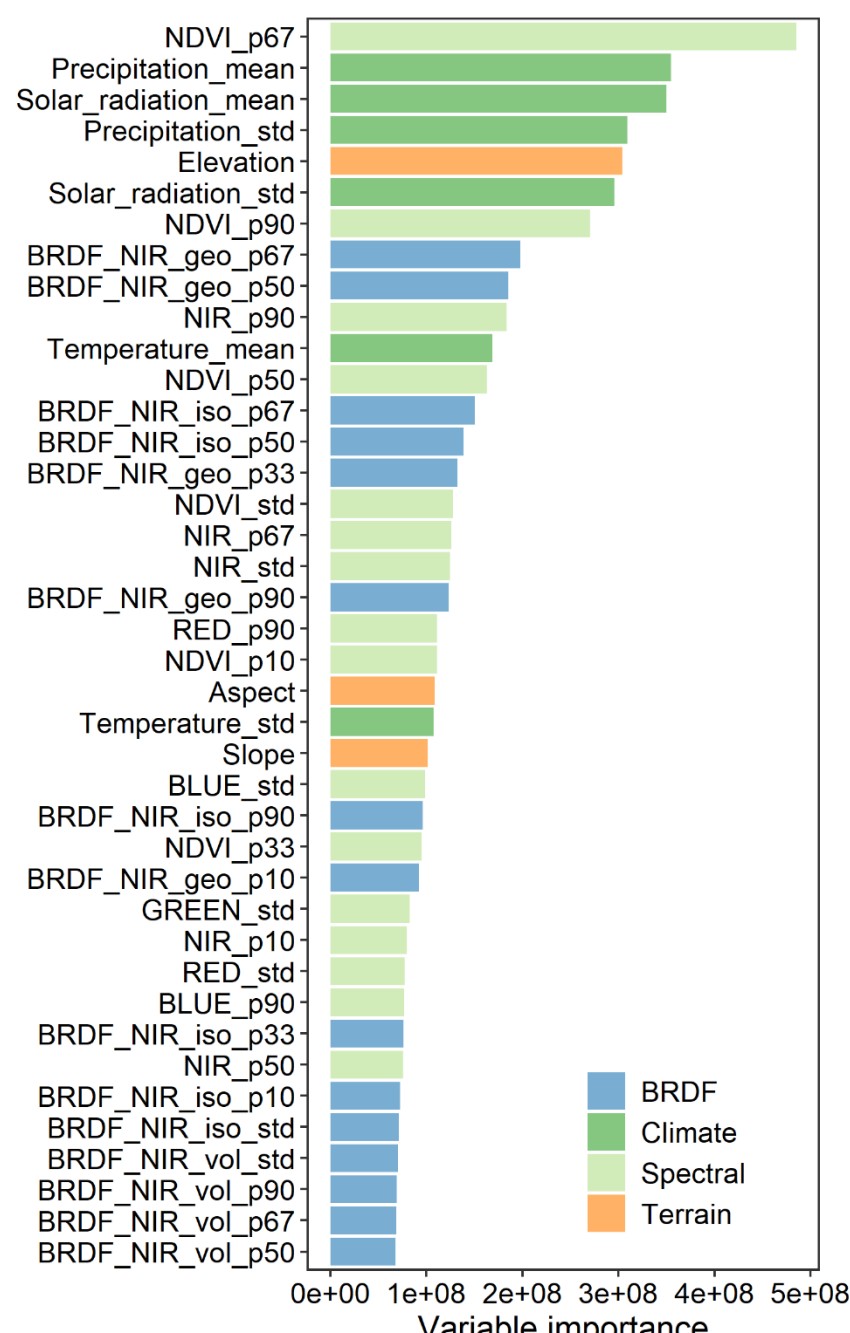

**Figure 6.** The importance of variables in the mean leaf inclination angle prediction. NIR, Red, Green, and Blue denote the nadir reflectance in near-infrared, red, green, and blue bands, respectively; geo, iso, and vol represent kernel coefficients of geometric-optical surface scattering, isotropic scattering, and volumetric scattering, respectively. The suffixes p××, mean, and std represent ××% quantile, mean, and standard deviation, respectively.

Fig. 7 illustrates the relationships between the upscaled MLA samples and the 16 most important variables. Overall, MLA
decreases with the increase of NDVI, NIR reflectance, and NIR BRDF kernel parameters, whereas it increases with the
standard deviation of NDVI. MLA is negatively correlated with solar radiation, precipitation, and temperature. Additionally,
MLA increases with increasing the standard deviation of solar radiation (corresponding to mid-to-high latitude regions),
while it decreases with the increase in the standard deviation of precipitation (corresponding to tropical and subtropical
regions with high precipitation). MLA increases slightly with altitude and then decreases.

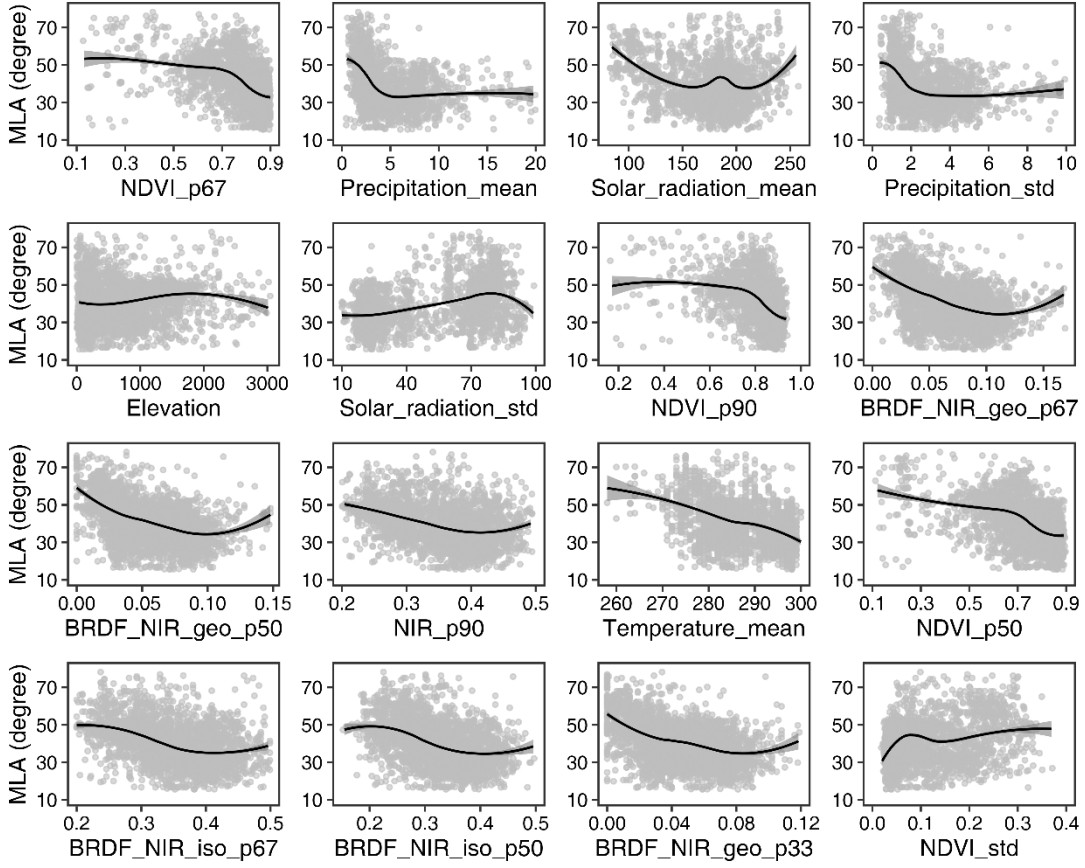


**Figure 7.** Relationships between mean leaf inclination angle (MLA) and different predictive variables. See Fig. 6 for different variables.

**3.3 Global MLA and G(0) maps**
Fig. 8 shows the spatial distribution of the global 500 m MLA product. Central Asia (grasslands), southern India (cereal
crops), and the central United States (grasslands and cereal crops) show higher MLAs of approximately 60°, whereas the
rainforests and Southeast Asia forests have more horizontal leaves with MLAs of around 30° (Fig. 8 and S2). MLA increases
with latitude, from $32.93 \pm 7.03°$ around the equator (~1.5° N) to $53.48 \pm 3.20°$ in the northern tundra (~76.5° N). Variation
in MLA decreases as latitude increases (Fig. 8). Among different PFTs, cereal crops show the highest MLA ($54.65 \pm 6.28°$),

while evergreen broadleaf forest has the lowest MLA (34.40 ± 6.42°), and PFTs follow the order: CRO-C > CRO-B > DNF > SHR > ENF ≈ GRA > DBF > EBF (Table 4). Grassland, broadleaf forest, and evergreen needleleaf forests show larger MLA variations than other PFTs, whereas deciduous needleleaf forests show minimal variation. The global vegetation MLA is 41.47°, with a standard deviation of 9.55°, which is comparable to the MLA of DBF (41.23 ± 6.58°) (Fig. 9a and Table 4).

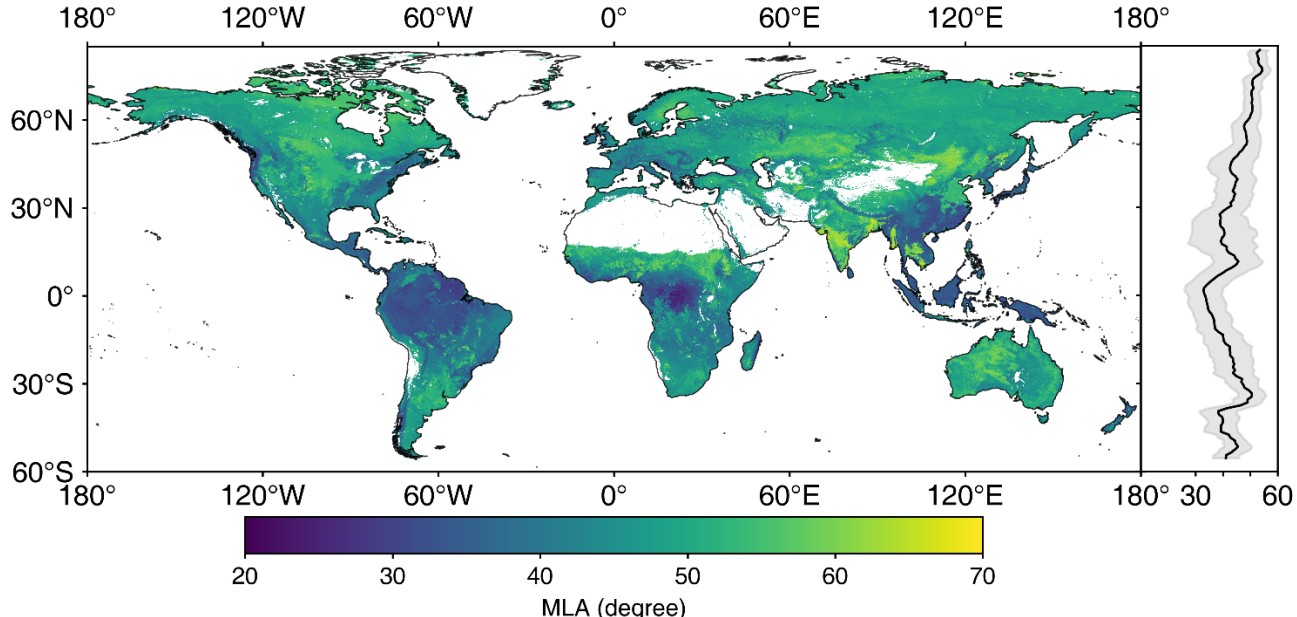

**Figure 8.** The global mean leaf inclination angle (MLA) map. The right panel shows the MLA latitudinal mean (solid line) and the standard deviation values (shaded area) weighted by leaf area index.

**Table 4.** Statistics of global mean leaf inclination angle (MLA), nadir leaf projection function (G(0)), and inclination index ($\chi_L$) for different plant functional types (PFT). STD is the standard deviation. The $\chi_L$ is converted from MLA ($\chi_L = 2cos(\text{MLA}) - 1$) (Lawrence et al., 2019).

| PFT | DBF | DNF | EBF | ENF | CRO-B | CRO-C | GRA | SHR | Globe |
|---|---|---|---|---|---|---|---|---|---|
| Area proportion(%) | 14.02 | 6.32 | 15.08 | 11.42 | 2.99 | 6.84 | 28.45 | 14.88 | 100.00 |
| MLA(°) | 41.23 | 50.05 | **34.40** | 47.13 | 52.35 | **54.65** | 47.12 | 49.23 | 41.47 |
| STD of MLA (°) | 6.58 | 3.24 | 6.42 | 8.35 | 6.63 | 6.28 | 8.08 | 5.35 | 9.55 |
| G(0) | 0.69 | 0.58 | **0.76** | 0.61 | 0.55 | **0.52** | 0.61 | 0.59 | 0.68 |
| STD of G(0) | 0.07 | 0.03 | 0.06 | 0.08 | 0.07 | 0.08 | 0.09 | 0.06 | 0.11 |
| $\chi_L$ | 0.50 | 0.28 | 0.65 | 0.36 | 0.22 | 0.16 | 0.36 | 0.31 | 0.50 |

The global MLA exhibits an asymmetric probability density distribution toward the lower MLA (Fig. 9b). It roughly presents three peaks, with the highest peak (~51°) containing DNF, ENF, CRO, GRA, and SHR. The moderate peak (~35°) is mainly composed of EBF and DBF, while the third peak (~58°) is dominated by crops. The MLAs of crops and some grasslands are close to the MLA of the spherical distribution (57.30°). The global MLA (41.47°) is 15.83° (38%) smaller than the MLA of the spherical distribution because the vegetation MLA is mostly less than 57.30° (Fig. 9b).

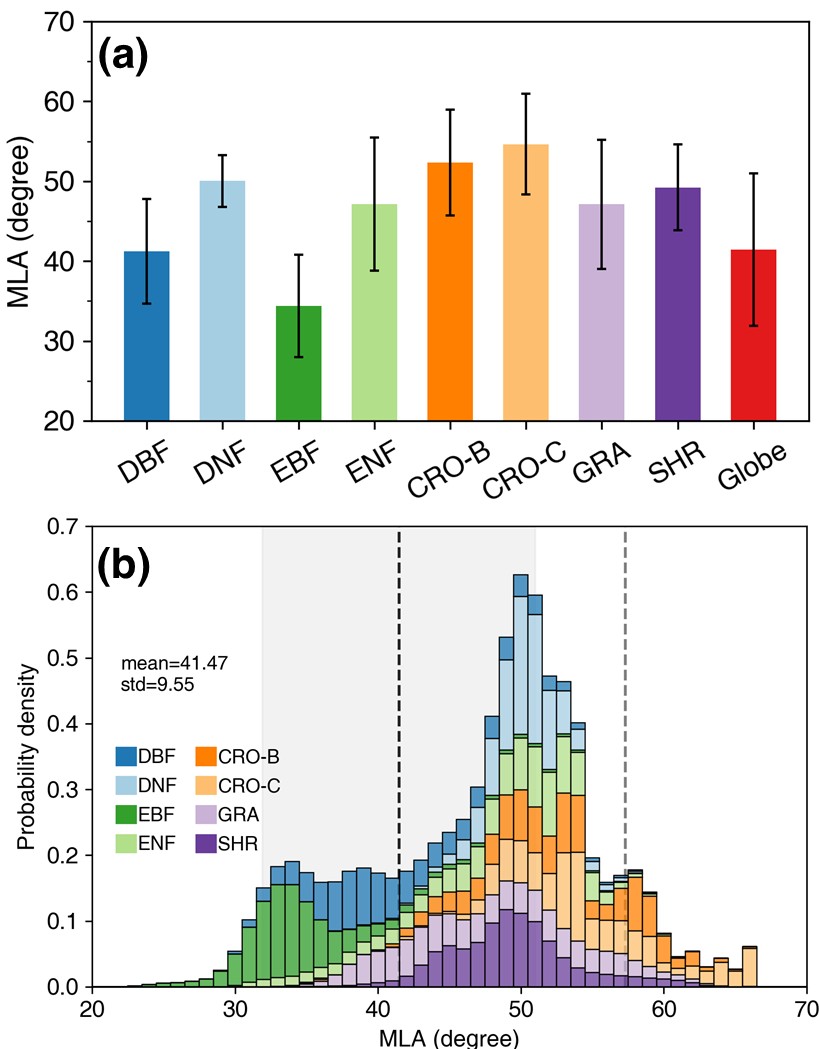

**Figure 9.** Statistics (a) and probability density distributions (b) of the global mean leaf inclination angle (MLA) for different plant functional types. The error bars in (a) represent the standard deviation. The black dash line and shade area in (b) indicate the global MLA mean and standard deviation. The gray dashed line represents the MLA (=57.30°) of spherical leaf angle distribution. The mean, standard deviation, and probability density values are weighted by leaf area index. See Fig. 1 for the acronyms.

Fig. 10 displays the spatial distribution of global G(0) generated from MLA. Overall, the global G(0) shows an opposite pattern with the global MLA. The G(0) values in Central Asia (grasslands, Fig. S2), southern India (cereal crops), and the central United States (grasslands and cereal crops) are relatively lower than those in tropical rainforests, forests in Southeast Asia, and forests in the eastern United States. G(0) generally decreases slowly with latitude, from $0.78 \pm 0.08$ at the equator (~1.5° N) to $0.52 \pm 0.04$ in the northern tundra (~76.5° N).

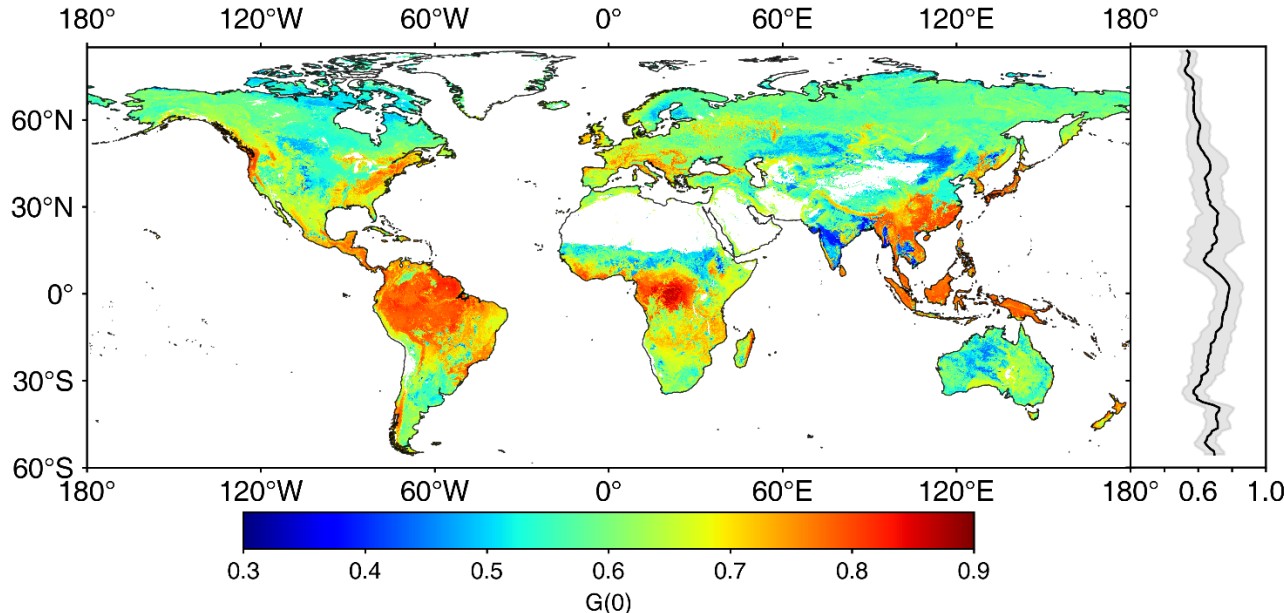


**Figure 10.** The global nadir leaf projection function (G(0)) map. The right panel shows the G(0) mean (solid line) and standard deviation

values (shaded area) weighted by leaf area index.
Among different PFTs, EBF has the highest G(0), at approximately 0.76 ± 0.06 (Fig. 11a, Table 4), whereas cereal crops
show the lowest value, at approximately 0.52 ± 0.08. The DBF G(0) is comparable to the global average. The G(0) of broad-
leaved forests is greater than that of other PFTs (Fig. 11a, Table 4). The global G(0) probability density distribution peaks at
0.52–0.65, with an asymmetric distribution (Fig. 11b). The proportion on the right side of the peak is larger than that on the
left. The peak of the global G(0) distribution mainly contains DNF, ENF, CRO, GRA, and SHR. The left side of the peak is
mainly composed of crops, while the right side is dominated by broad-leaved forests and some shrubs. The spherical
distribution G(0) (0.50) is mainly represented by crops and a small amount of grassland, where G(0) also shows a large
variation (~0.35). The spherical distribution G(0) is 0.18 (26%) less than the global average G(0) (0.68), as most vegetation
G(0) is greater than 0.50 (Fig. 11b).

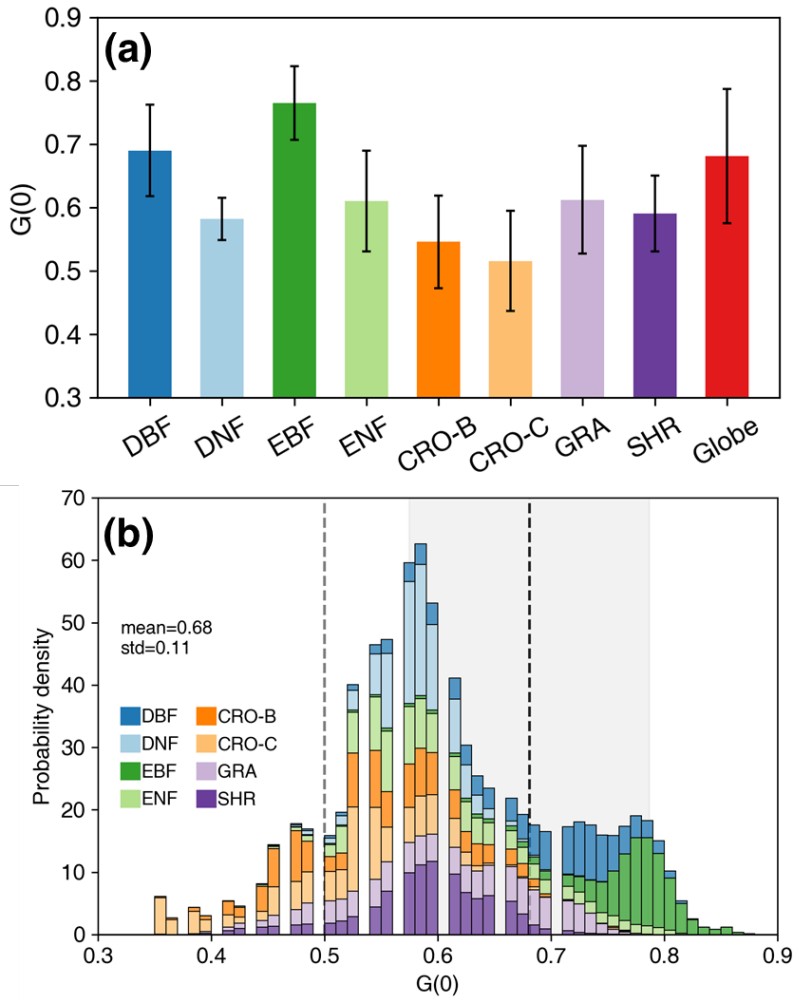


**Figure 11.** Statistics (a) and probability density distributions (b) of the global nadir leaf projection function (G(0)) for different plant functional types. The error bars in (a) represent the standard deviation. The black dash line and shade area in (b) indicate the global G(0) mean and standard deviation. The gray dashed line represents the G(0) (=0.50) of spherical leaf angle distribution. The mean, standard deviation, and probability density values are weighted by leaf area index. See Fig. 1 for the acronyms.

Fig. 12 demonstrates the global distributions of the MLA quality indicators. The global mean proportion of high-quality BRDF inputs is 68.03%. Northern South America and Central Africa have a low proportion of high-quality inputs (20%) because of cloud contamination (Fig. 12 (a)). Considering the large number of observations for each pixel (7904 from 2001 to 2022), this percentage (20%) of high-quality observations is sufficient to map MLA. In addition, 80.39% of the global MLA map was derived within the feature ranges of training samples, and the rest 19.61% were mainly located in high-latitude regions and Africa. For the latter areas, the MLA map was predicted with extrapolation and caution should be taken when using the map (Fig. 12 (b)).

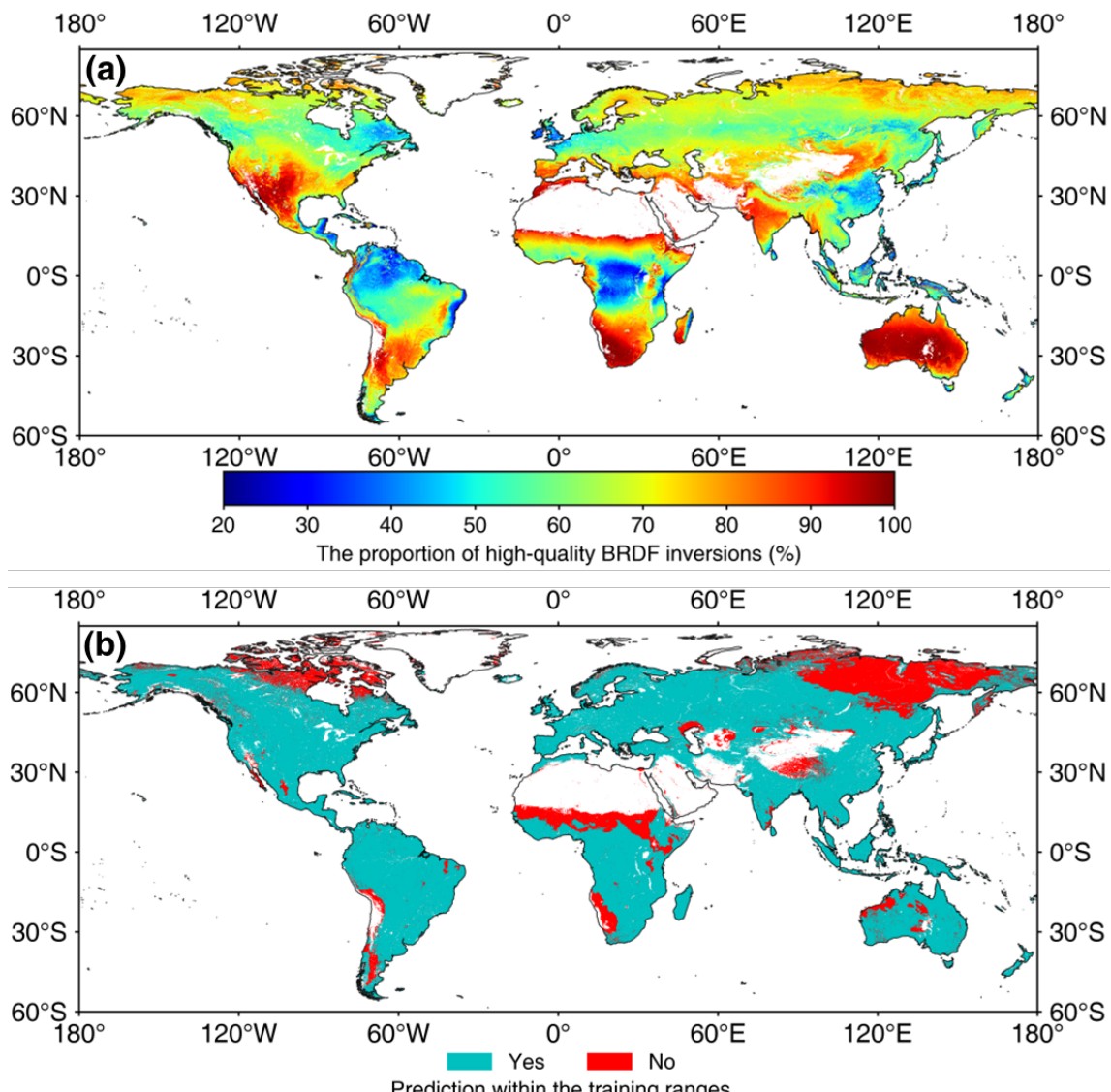

**Figure 12.** Global distributions of quality indicators. (a) and (b) denote the proportion of high-quality BRDF inversions, and whether the prediction is within the ranges of training samples, respectively.

## 3.4 Evaluation of global MLA

Fig. 13 shows the comparison between the predicted MLA and upscaled MLA samples using the ten-fold cross-validation method. For noncrops, the predicted MLA is moderately consistent with the upscaled sample MLA ($r = 0.75$, RMSE = 7.15°), with 83% of samples having residuals < 10° and 94% of samples having residuals < 15°. For DNF and SHR, the

predicted MLA compresses the variation range of sample MLA (Fig. 13a). For crops, the predicted MLA of CRO-C shows
higher consistency ($r = 0.60$) than that of CRO-B ($r = 0.48$). (Fig. 13b and c).

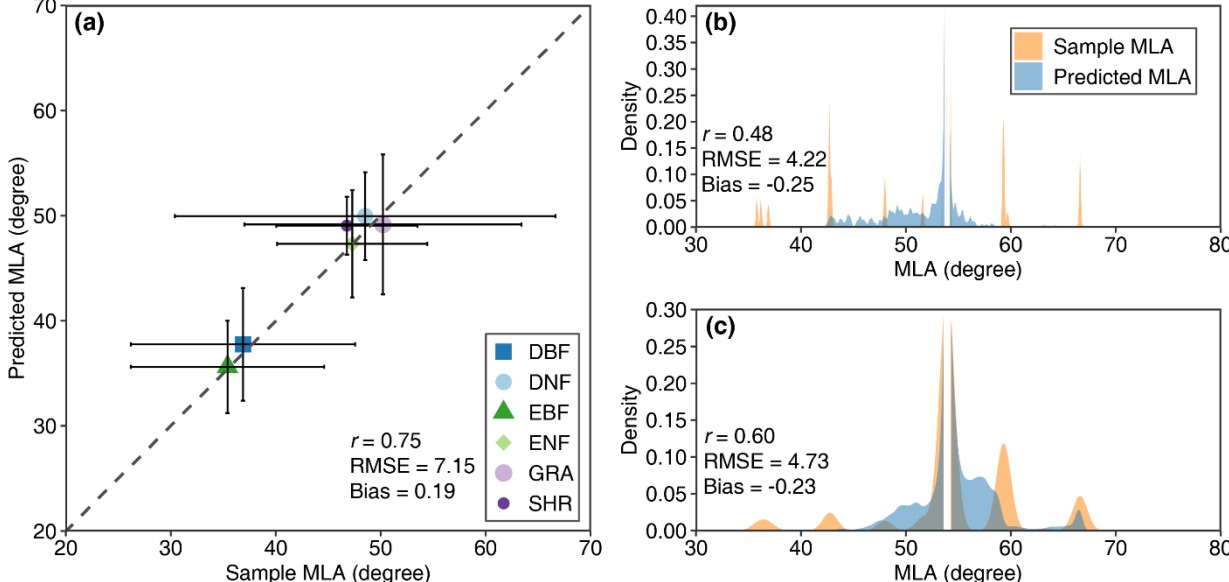


**Figure 13.** Comparisons between predicted MLA and sample MLA for noncrop (a), broadleaf crops (b), and cereal crops (c) (See Fig. 1
for the acronyms). The error bar in (a) represents the standard deviation.
Fig. 14 compares G(0) derived from the MLA and high-resolution reference data. The MLA-derived G(0) shows moderate
consistency with the reference G(0) ($r = 0.62$), and 65% of the estimated G(0) residuals are < 0.15, and 84% of the residuals
are < 0.20. The estimated G(0) generally overestimates (bias = 0.11), especially when G(0) is low (< 0.60), mainly for crops,
pasture, woody wetlands, and shrubs, whereas grasslands show better consistency. The estimated G(0) is temporally more
stable than the reference G(0) which is generally greater than 0.50 and displays seasonal variation (horizontally distributed
bars in Fig. 14).

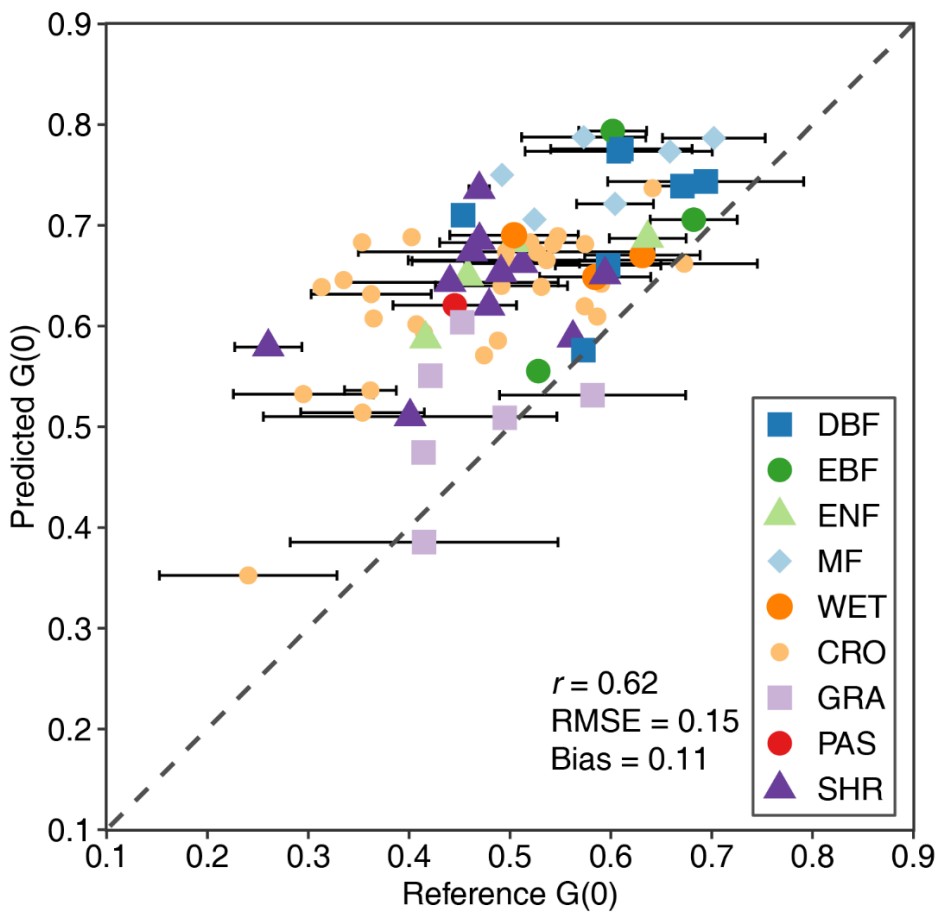


**Figure 14.** Comparisons of G(0) derived from mean leaf inclination angle and high-resolution reference data for different plant functional
types (see Fig. 2 for the acronyms). The error bar represents the standard deviation of reference G(0) at different seasons.
**4 Discussion**
**4.1 Global MLA and G(0)**
This study compiled global LIA field measurements and generated the first global 500 m MLA and G(0) maps (Figs. 8 and
10). These maps show the average MLA and G(0) conditions during the growing seasons from 2001 to 2022. Overall, the
global MLA is lowest around the equator and increases with latitude (Figs. 8 and 10). This accords with the MLA latitude
variation derived from model simulations (Huemmrich, 2013). Crops have higher MLA than broadleaf forests whose leaves
are relatively horizontal. The global MLA and G(0) maps enhance our understanding of the global distribution of MLA and
G(0) and should be useful in radiative transfer modeling, remote sensing of vegetation parameters, land surface modeling,
and ecological studies.

The global MLA shows good consistency with validation samples (Fig. 13) and the statistics of LIA field measurements (Tables 3 and 4), demonstrating its reliability. The globally derived MLA is 41.47°, which is consistent with the LIA measurements (40.74°, Tables 3 and 4). However, the derived MLAs of DBF, DNF, CRO-B, and SHR are approximately 10° greater than the measured MLAs. It is noted that the number and spatial distribution of LIA measurements for these biomes are limited. For example, the global CRO-B areas are dominated by soybeans with higher LIA (Table S2), and the LIA measurements for soybeans are limited, which caused the CRO-B MLA in the global map to be greater than that in the measurement statistics (Tables 3 and 4). The poor crop MLA prediction (Fig. 13b) is mainly caused by a small number of samples and the strong seasonal variation. It is difficult to consider within-crop LIA variation when typical MLA values are assigned to different crops.

Due to the lack of high-resolution reference MLA, the global MLA was evaluated through a comparison of the MLA-derived $G(0)$ with the high-resolution reference $G(0)$ (Fig. 14). This practice was adopted because both MLA and $G(0)$ are closely related. $G(0)$ is typically calculated from the LIA distribution function based on Nilson's algorithm (Nilson, 1971). We calculated $G(0)$ from MLA assuming an ellipsoidal LIA distribution (De Wit, 1965) and found that the calculated $G(0)$ is highly consistent with the reference $G(0)$ calculated from the Nilson's algorithm for different theoretical LIA distributions (Fig. S5). The MLA-calculated $G(0)$ also shows a monotonic decreasing relationship with MLA (Fig. S6).

The result shows medium consistency but MLA-derived $G(0)$ overestimates at low values (< 0.60), especially for CRO, PAS, SHR, and WET. The overestimation may be partly caused by the underestimation of MLA at high values that is related to the errors introduced in the sample expansion and upscaling. These errors are mainly caused by a lack of LIA measurements, vegetation structural complexity, and seasonal variation. In addition, the uncertainties in the reference $G(0)$ may have contributed to the overestimation. The reference $G(0)$ was derived from the Beer-Lambert law (Eq. (4)) which assumes that the canopy is a turbid medium. The turbid medium assumption is unrealistic for complex vegetation (Widlowski et al., 2014). The angular variation of CI and the mixture of branches and leaves in generating high-resolution $G(0)$ can also lead to the overestimation. Previous studies have shown that CI increases with the view zenith angle (Fang, 2021), which means that the whole CI > CI(0) and can lead to the underestimation of the reference $G(0)$ (Eq. (6) and (7)). The mixture of branches and leaves may result in the underestimation of the reference $G(0)$ due to the usually higher inclination angle of the trunks (Liu et al., 2019). The MODIS LAI product used for LIA upscaling in the $G(0)$ validation (section 2.4) is known to have issues such as internal inconsistency, backup algorithm accuracy, and spatiotemporal gaps (Kandasamy et al., 2013; Pu et al., 2023; Zhang et al., 2024). In the future, new improved MODIS LAI can be used in the $G(0)$ validation (Pu et al., 2024; Yan et al., 2024). Compared with the previous $G(0)$ derived from global vegetation biophysical products (Eq. (7)) ($R^2 = 0.11$, RMSE = 0.53) (Li et al., 2022), the MLA-derived $G(0)$ performs better (R = 0.62, RMSE = 0.15). In addition, the $G(0)$ data obtained from our study can be used to derive the $G(\theta)$ for any arbitrary angle. One method of getting $G(\theta)$ is based on single-parameter ellipsoidal leaf angle distribution (Campbell, 1990) (Eq. (3)). Another method is to make use of both $G(0)$ and $G(57.3°)$ ($\equiv 0.5$) and derive $G(\theta)$ using a simple linear ($G(\theta) = a \cdot \theta + b$) or sinusoidal ($G(\theta) = a \cdot sin(\theta) + b$))

interpolation method. Since G(θ) varies most significantly in the nadir direction for different MLA (Wilson, 1959), the
uncertainty of G(θ) derived from the global MLA in other directions will be smaller than that of G(0).

## 411     4.2 The relationship between MLA and other variables

Analysis of the relationships between MLA and other features in the MLA mapping process reveals that MLA is negatively
correlated with NDVI, NIR reflectance, and NIR BRDF kernel coefficients (Fig. 7). These findings are consistent with other
simulation and experimental studies (Zou and Mõttus, 2015; Liu et al., 2012; Dong et al., 2019; Jacquemoud et al., 1994).
Higher MLA means lower radiation interception, more NIR and red downward radiations reach the soil background. This
causes lower NIR and higher red reflectance because the soil background typically has lower (higher) reflectance for NIR
(red) (Siegmund and Menz, 2005). This results in negative correlations between MLA and NIR reflectance and NDVI (Liu et
al., 2012). The negative relationships between MLA and radiation, precipitation, and temperature (Fig. 7) are related to the
vegetation adaptation mechanism. Under suitable climate conditions (radiation, precipitation, and temperature), horizontal
leaves are formed to absorb more radiation and increase the photosynthesis rate (Van Zanten et al., 2010; King, 1997). The
positive correlation between MLA and the standard deviation of radiation and temperature (Fig. 7) indicates that the MLA is
more vertical in areas with significant seasonal changes in radiation and temperature (mid to high-latitude areas) because
vertical leaves maximize intercepted radiation under low solar altitudes at mid to high-latitude areas (Huemmrich, 2013).
Plant function type was initially used as a predictive variable (Tables 1 and 2), but relatively low importance was found for
LIA prediction (Fig.6, ranked 47 out of 76). This may be because the biome information is implicitly included in the spectral
features as the former is frequently derived from the latter (Sulla-Menashe et al., 2019). Previous studies have demonstrated
that the LIA variation within PFTs may be larger than that between PFTs. This indicates that the PFT is not a good predictor
(Prentice et al., 2024). To avoid overfitting, only the most important 40 features were used for MLA prediction (Fig. 6). To
explore the regional differences of the variable importance, an analysis was conducted for the tropical (23.5°S-23.5°N),
northern temperate (23.5°N-60°N), northern polar (60°N-90°N), and the southern temperate (23.5°S-60°S) zones. The 40
most important variables are similar among different regions although minor differences exist (Fig. S7). Among the 40
variables for tropical, northern temperate, northern polar, and southern temperate zones, 32, 35, 30, and 31 of them,
respectively, are the same as the 40 global variables (Fig. S7). Climate and spectral variables are significant among all
regions, whereas BRDF features are the most important in the southern temperate zone. The 40 most important variables in
the global MLA prediction account for ~ 80% of total importance among different regions, which is similar to that in the
global prediction.

## 437     4.3 Use of the new MLA map

The spherical LAD assumption has been widely adopted in the literature (Tang et al., 2016; Zhao et al., 2020; Wang and
Fang, 2020). This study demonstrates that the spherical assumption is valid only for cereal crops, but not for broadleaf

forests (Tables 3 and 4). This finding is consistent with previous local LIA measurements (De Wit, 1965; Pisek et al., 2013; Yan et al., 2021). For crops, the spherical assumption may even become invalid because of seasonality and species diversity (Table S2, Figs. 5 and 9). Fig. 14 shows that most of the reference G(0) values are greater than 0.50, while the spherical distribution would underestimate the interception of radiation and rainfall (Figs. 9 and 11) (Stadt and Lieffers, 2000). In current LSMs, a constant LIA is commonly assigned for each PFT (Majasalmi and Bright, 2019). For example, the Community Land Model V5 (CLM5) (Table S4) (Lawrence et al., 2019) uses lower inclination indices and higher LIA values than our results (Tables 3 and 4) and thus may underestimate canopy interception. The global LIA map generated in this study provides a more reasonable LIA parameterization strategy for the application communities.

**4.4 Limitations and prospects**

The limitations of this study mainly relate to the small number of LIA measurements, especially continuous measurements. First, within-species LIA variations were neglected in the spatial expansion due to limited spatial coverage of existing LIA-measured data (Section 2.3.1). This may introduce some errors, especially for crops. Second, three different sources of LIA measurements were gathered with different measurement schemes, and uncertainty may arise because of these differences. The random forest algorithm is robust to these differences because part of the samples and features were randomly selected and the algorithm ensembled the predictions from multiple decision trees (Svetnik et al., 2003). We manually inspected all field LIA data and are confident in their data quality. Third, for forests, the contribution of the understory was not considered. Typically, the understory is characterized by more horizontal leaves, and ignoring the understory may lead to an MLA overestimation (Utsugi et al., 2006). Nevertheless, a previous study showed that the relative contribution of the understory to the overall MLA is less than 10% (Li et al., 2022). Finally, only the growing season MLA was calculated, whereas the seasonal and long-term variations of MLA were not considered due to the lack of continuous LIA measurements.

We assumed a linear LAI-EVI2 relationship (LAI = a*EVI2) to upscale MLA from the canopy to 500 m scale (Section 2.3.1 and Appendix A). Global analysis of MODIS LAI and EVI2 products shows a slight non-linear relationship between them (Fig. S8). The non-linear relationship was also used to upscale MLA (Eq. A2) in a side experiment, where the derived MLA was found consistent with the original one (Fig. S9) because of the homogeneity of the 500 m pixel after rigorous sample screening (section 2.3.1). This demonstrates the suitability of the linear assumption.

In the future, more efficient LIA observation systems should be developed to provide continuous LIA data (Kattenborn et al., 2022). LIA measurements can be integrated into existing ground observation networks, such as the National Ecological Observatory Network (NEON) (Kao et al., 2012), Integrated Carbon Observation System (ICOS) (Gielen et al., 2018), and Terrestrial Ecosystem Research Network (TERN) (Karan et al., 2016), to enhance temporal LIA measurements in larger spatial extent, especially for DNF and crops. Using standard LIA measurement protocols will certainly improve the LIA data consistency. (Li et al., 2023). In addition, canopy structure parameters are interrelated, and introducing other structure parameter products, such as LAI, FVC, CI, and canopy height as predictive variables may improve the MLA prediction. Multiangle reflectance (Jacquemoud et al., 2009; Goel and Thompson, 1984; Jacquemoud et al., 1994) or light detection and

ranging (Zheng and Moskal, 2012; Bailey and Mahaffee, 2017; Itakura and Hosoi, 2019) are encouraging remote sensing
tools that can help to derive temporally continuous and high-resolution MLA data.
**5 Conclusion**
This study compiled existing global LIA measurements and generated the first global 500 m MLA and G(0) products by
gap-filling the LIA measurement data using a random forest regressor. The mean of global LIA measurements is 40.74° and
cereal crops show the highest MLA (59.11°). The global MLA shows an explicit spatial distribution and the value increases
with latitude. The global MLA is $41.47° \pm 9.55°$ and follows the order of CRO-C > CRO-B > DNF > SHR > ENF $\approx$ GRA >
DBF > EBF. The predicted MLA presents a medium consistency ($r = 0.75$, RMSE = 7.15°) with the validation samples for
noncrops. For crops, the results are relatively poorer ($r = 0.48$ and 0.60 for broadleaf crops and cereal crops) because of
limited LIA measurements and strong seasonality. The G(0) derived from MLA is moderately consistent with the reference
G(0) ($r = 0.62$).
The MLA and G(0) products obtained in this study would enhance our understanding of global LIA and assist remote
sensing retrieval and land surface modeling studies. These products provide a more realistic parameterization strategy than
the commonly used spherical LAD and PFT-specific MLA assignment. Note the global MLA and G(0) products mainly
represent the typical state during the growing season. These products can be further improved and temporal MLA data can
be obtained through continuous measurements and remote sensing retrieval.
**Data availability**
The global MLA and G(0) products (CAS-GLA) are available in: Li, S. and Fang, H. 2024,
https://doi.org/10.5281/zenodo.12739662(Li and Fang, 2025). The related code can be accessed at
https://code.earthengine.google.com/?accept_repo=users/SiJia/MTA.
**Author contributions**
HF and SL conceptualized this work. SL compiled global LIA measurements, generated global products, and curated the
datasets. SL and HF wrote the manuscript. HF was responsible for funding and supervision.
**Competing interests**
The contact author has declared that none of the authors has any competing interests.

**Acknowledgements**

The authors are grateful to TRY and many other researchers for sharing the LIA measurement data. Jens Kattge at the Max Planck Institute for Biogeochemistry and Dongliang Cheng at Fujian Normal University provided the TRY species location data and LIA measurements in China's subtropical regions, respectively.

**Financial support**

This work was mainly supported by the National Natural Science Foundation of China (42171358).

**Appendix A. Upscaling LIA from leaf, canopy to ecosystem scale**

From leaf to canopy scale, the entire canopy MLA is commonly calculated as the average of all measured leaf LIAs weighted by leaf area (Eq. A1) ([Zou et al., 2014](); [De Wit, 1965](); [Yan et al., 2021]()). In practice, because of the difficulty in leaf area measurement, especially for a large number of leaves, the variability of leaf areas within a canopy is often ignored and the areas of all leaves are assumed similar. In this case, the canopy LIA can be simplified as the average LIA weighted by leaf number (Eq. A1) ([Ryu et al., 2010](); [Pisek et al., 2011](); [Chianucci et al., 2018]()):

$$MLA_{canopy} = \frac{\sum_i LIA_i * LA_i}{\sum_i LA_i} = \frac{LA_{mean} * \sum_i LIA_i}{LA_{mean} * N} = \frac{\sum_i LIA_i}{N} \tag{A1}$$

where $MLA_{canopy}$ is the MLA at canopy scale, $i$ is the $i$th leaf, LIA is leaf inclination angle, LA is single leaf area, $LA_{mean}$ is the mean leaf area by ignoring the variation of leaf area within a canopy, N is number of leaves within a canopy.

From the canopy to 30 m scale, the canopy level MLA is regarded as equal to 30 m-MLA because for MLA measurements, the dominant species was artificially identified by investigators and the spatial representativeness at the extent of 30 m is ensured.

From 30 m to 500 m, the 500 m MLA was formulated as the weighted average of 30 m MLA by the leaf area of the 30 m pixel (Eq. A2), the same as that from the leaf to canopy scale. The leaf area of a 30 m pixel can be deduced from the product of leaf area index (LAI) and the ground area of a 30 m pixel according to the definition of LAI (the half of green leaf area on the unit of ground area) (Eq. A2) ([Fang et al., 2019]()).

$$MLA_{500} = \frac{\sum_j MLA_{30\_j} * LA_{30\_j}}{\sum_j LA_{30\_j}} = \frac{\sum_j MLA_{30\_j} * LAI_{30\_j} * S}{\sum_j LAI_{30\_j} * S} = \frac{\sum_j MLA_{30\_j} * LAI_{30\_j}}{\sum_j LAI_{30\_j}} \tag{A2}$$

Where $MLA_{500}$ and $MLA_{30}$ represent MLA at 500 m and 30 m scales, $j$ is the $j$th 30 m pixel, $LA_{30\_j}$ is the total leaf area of a 30 m pixel, $LAI_{30\_j}$ is leaf area index (m2/m2) of a 30 m pixel, $S$ is the ground area of a 30 m pixel.

Assuming LAI = a*EVI2+b and b ≈ 0 (as illustrated in Fig. S8), the MLA at 500 m scale can be calculated as:

524     $MLA_{500} = \frac{\sum_j MLA_{30\_j} * EVI2_{30\_j}}{\sum_j EVI2_{30\_j}}$                                                                    (A3)

525

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
