# Peer review of "Mapping global leaf inclination angle (LIA) based on field"

_Earth System Science Data, 2024_

## Author Comment (AC1)

**Referee #1**

This manuscript describes an effort to make a global reference map of leaf inclination angle by combining leaf angle observations (from the TRY database and extracted from images) with ancillary data (including plant functional/crop types, reflectance, BRDF, climate, topography) and a random forest approach. Results are compared to other available data related to leaf angle distributions from the GBOV and DIRECT databases.

The clearly written manuscript provides a compelling justification for why consistent global leaf angle data would be widely useful. The authors note the challenge of sparse leaf angle observations, and while they have devised some creative ways to expand those observations to train the random forest model, some elements of the methods and evaluation have the potential to create consequential bias.

We thank the referee for the recognition and insightful comments that help us improve the manuscript. We have noted the biases in Fig. 13 and discussed their causes in section 4.1 (Line 346-359).

> *Due to the lack of high-resolution reference MLA, the global MLA was evaluated through a comparison of the MLA-derived $G(0)$ with the high-resolution reference $G(0)$ (Fig. 13). The result shows medium consistency but MLA-derived $G(0)$ overestimates at low values (< 0.60), especially for CRO, PAS, SHR, and WET.* **The overestimation may be partly caused by the underestimation of MLA at high values that is related to the errors introduced in the sample expansion and upscaling. These errors are mainly caused by a lack of LIA measurements, vegetation structural complexity, and seasonal variation. In addition, the uncertainties in the reference $G(0)$ may have contributed to the overestimation.** *The reference $G(0)$ was derived from the Beer-Lambert law (Eq. (4)) which assumes that the canopy is a* **turbid medium.** *The turbid medium assumption is unrealistic for complex vegetation (Widlowski et al., 2014).* **The angular variation of CI and the mixture of branches and leaves** *in generating high-resolution $G(0)$ can also lead to the overestimation. Previous studies have shown that CI increases with the view zenith angle (Fang 2021), which means that the whole $CI > CI(0)$ and can lead to the underestimation of the reference $G(0)$ (Eq. (6) and (7)). The mixture of branches and leaves may result in the underestimation of the reference $G(0)$ due to the usually higher inclination angle of the trunks (Liu et al. 2019b). Compared with the previous $G(0)$ derived from global vegetation biophysical products (Eq. (7)) ($R2 = 0.11$, RMSE = 0.53) (Li et al. 2022), the MLA-derived*

*G(0) performs better (R2 = 0.38, RMSE = 0.15).*

In addition, Since $G(\theta)$ varies most significantly in the nadir direction for different

MLA (Wilson 1959), the uncertainty of $G(\theta)$ derived from the global MLA in other directions is smaller than that of $G(0)$.

Specific comments:

1.    The method from Pisek et al. (2011) to derive leaf angle from images requires that images are leveled. It's not possible to know whether images taken from Google are leveled, and whether images systematically describe distribution within a plant, and this can create bias in the dataset.

The referee is correct that the canopy pictures taken from Google do not contain the level information directly. In this study, the level state of the canopy images was determined from the background information, such as the ground level and plant stems.

For each species, more than 75 leaves from different images were collected (Line 110), reducing the uncertainties from non-leveled photography.

2.    The TRY database was used to determine dominant species in an area to select species for manual classification from images. No details were given about how this was done, but datasets from TRY were not designed for this purpose and may not be representative.

Thanks to the referee's reminder, we have added more details regarding the species selection procedure to the manuscript (Line 108).

*The TRY species location data (848,919, Fig. S3b) (Jan 03, 2022) were used to*

*obtain the dominant species information in tropical rainforests and the northern*

*tundra. The species location points in these two vegetation types were spatially*

*filtered and the frequency of occurrence for each species was counted. The species*

*with a high frequency of occurrence were selected to measure the LIA.*

Most species distribution databases, e.g., the Global Biodiversity Information Facility (GBIF) (Yesson et al. 2007), only consider the appearance of species but not their spatial representativeness. The TRY species location database consists of trait measurements for common species which represent a hundreds-of-square-meters area around the location. The dominant species was artificially identified by investigators and the spatial representativeness is vital for following LIA upscaling. Therefore, the
TRY species location database was utilized after throughout consideration.
3.   Leaf angle can be highly variable within a species, depending on factors like leaf
age, plant water status, and canopy position. The manuscript does not report
distributions of replicates per species, and given the large expansion of spatial coverage
from TRY data locations (where leaf angles were not directly observed) it's possible
that training data may not be representative of their species.
We agree with the referee about the leaf angle variation from a plant physiological
perspective. It is understood that LIA is influenced by the environment and varies
within a species.
In this study, LIA is the mean leaf inclination angle (MLA) of all leaves at the canopy
or pixel scale, not for a single leaf. For a site, the LIA of multiple leaves at different
heights and orientations are obtained and averaged to obtain a robust MLA (Chianucci
et al. 2018; Pisek and Adamson 2020). The MLA partly mitigates the impact of canopy
position, sunlit and shaded leaves, branching patterns, stem elongation, and species-
specific genetic traits like phototropism and heliotropism. This kind of mean LIA is
desperately wanted in many remote sensing and land surface modeling studies
(Lawrence et al. 2019; Li et al. 2023; Majasalmi and Bright 2019; Tang et al. 2016;
Zhao et al. 2020). In those studies, LIA is commonly assumed constant (spherical
distribution, 57.3 degrees) or biome type-specific (assigning a constant value for each
biome). Indeed, these assumptions may not represent the true field measurements
(Tables 3 and 4). Our objective is to provide a more realistic global MLA map for
remote sensing and land surface modeling studies.
In this study, the LIA seasonal variations were not considered in the global LIA map
because of the lack of seasonal LIA measurements. As a matter of fact, temporal LIA
variations are usually small, except under extreme situations (unusual). For example,
the LIA variations of European beech forest and eucalyptus in different successional
stages are less than 10 degrees (le Maire et al. 2011; Liu et al. 2019; Raabe et al. 2015).
Crops generally show higher LIA variations than non-crops (Biskup et al. 2007; Zhang
et al. 2017). Therefore, many studies have considered LIA as a species-specific static
trait when there are no seasonal field measurements (Pisek et al. 2022; Raabe et al. 2015;
Toda et al. 2022).

The global LIA map derived in this study is consistent with field measurements (Tables 3 and 4). This is a significant improvement compared to existing static simplifications (Lawrence et al. 2019; Li et al. 2023; Majasalmi and Bright 2019; Tang et al. 2016; Zhao et al. 2020). In a forthcoming study, we plan to retrieve LIA from remote sensing and the temporal LIA variation will be considered.

Thanks to the referee's comment, we have revised the manuscript (Line 151).

*Many studies have treated LIA as a species-specific static trait and ignored within-species variations when LIA measurements are limited (Pisek et al., 2022; Toda et al., 2022; Raabe et al., 2015). Following the rationale, the spatial coverage of LIA measurements was expanded, and those records without location information were utilized (section 2.1.1).*

In addition, we counted the number of locations for different species and found the LIA replicates per species range from 1 to 330, and most replicates (98%) are less than 50. We added this information to the manuscript (Line 118).

4. Some of the products used for upscaling and evaluation themselves depend on assumptions about leaf angle, including MODIS LAI which was used to upscale the mean leaf angle data produced here to compare to GBOV and DIRECT data. I expect that GBOV and DIRECT LAI products also depend on leaf angle assumptions (as almost all methods of estimating LAI do).

In the MODIS LAI algorithm, a biome-specific static LIA was used as a priori (Myneni et al. 2002). The LIA is partly considered in the LAI retrieval algorithm and the MODIS LAI has been widely validated and shows good consistency (Brown et al. 2020; Yan et al. 2021). Therefore, it was used to upscale LIA in the evaluation procedure.

In GBOV and DIRECT, the high-resolution reference LAI is estimated by the empirical relationship between reflectance and LAI measurements. The LAI measurements were obtained with the Miller method (Eq. (1)) which does not require any leaf angluar information (https://gbov.land.copernicus.eu/products/).

$$LAI = 2 \sum_{i=1}^{n} -\overline{ln\ P(\theta_i)} \cos(\theta_i) \sin(\theta_i) d_{\theta_i} \tag{1}$$

Where $P(\theta_i)$ is the gap fraction value in viewing zenith ring $i$. Therefore, the GBOV and DIRECT data do not dependent on leaf angle assumptions.

Technical comments:

1. Line 10: I recommend "trait" instead of "parameter" here when discussing ecological
processes.

We have revised it.

2. Line 103: I was confused by the statement "The majority of existing LIA
measurements are located in the mid-latitudes of the Northern Hemisphere." Because
Figure 1 looks like a huge amount of data are in the American tropics?

Two different versions of TRY data (V5 and V6) were used and the V6 data provide a
large amount of LIA measurements in the Southern Hemisphere. The original sentence
was deleted.

3. Line 159: Coefficient of variation in reflectance or something else?

Yes, it represents the coefficient of variation in reflectance. We have revised it in the
manuscript.

4. Line 194: The single-parameter ellipsoidal leaf angle distribution seems like a big
assumption. Where there are data to test this, does it seem reasonable?

Compared to other leaf angle distribution models, the single-parameter ellipsoidal leaf
angle distribution is a relatively more accurate and simpler model and has been used in
many remote sensing studies (Campbell 1990; Kuusk 2001; Verhoef et al. 2007; Wang
et al. 2007). Therefore, the single-parameter ellipsoidal leaf angle distribution was also
used in this study and its parameter $\chi$, the ratio of the horizontal and vertical axes of an
ellipsoid, was first derived from MLA. We have rephrased the original sentence (Line
201).

*Assuming a single-parameter ellipsoidal leaf angle distribution (Campbell, 1990),*
*the parameter $\chi$, the ratio of the horizontal and vertical axes of an ellipsoid, was*
*first derived from MLA. Compared to other models, the single-parameter*
*ellipsoidal leaf angle distribution is a relatively more accurate and simpler model*
*and has been used in many remote sensing studies (Kuusk 2001; Verhoef et al.*
*2007; Wang et al. 2007).*

5. Figure 12: Are the distinct peaks in the reference data for different crops in panels b and c?

The distinct peaks in the reference sample data are caused by the MLA assignment manner and the homogeneity of cropland. The crop MLA samples were generated by assigning typical MLAs (Table S2) for different crops with high-resolution crop maps, followed by the upscaling (section 2.3.2 Line 188). In the upscaling, the homogeneity of cropland may result in low sample diversity and distinct peaks.

We have clarified it in Lines 180 and 188.

> Different mapping strategies were employed for noncrops and crops (Fig. 3b) considering the small number of valid crop samples (Fig. 4) and the lack of location information for most crop samples.
>
> For crops, the measured MLA values were averaged for different crop types as a typical MLA (Table S2). After assigning typical MLAs for different crops with high-resolution crop maps (Table 1), the high-resolution crop MLA were upscaled to 500 m as training samples (Eq. (1)).

**Reference**

Biskup, B., Scharr, H., Schurr, U., & Rascher, U. (2007). A stereo imaging system for measuring structural parameters of plant canopies. *Plant, Cell and Environment, 30*, 1299-1308

Brown, L.A., Meier, C., Morris, H., Pastor-Guzman, J., Bai, G., Lerebourg, C., Gobron, N., Lanconelli, C., Clerici, M., & Dash, J. (2020). Evaluation of global leaf area index and fraction of absorbed photosynthetically active radiation products over North America using Copernicus Ground Based Observations for Validation data. *Remote Sensing of Environment, 247*

Campbell, G. (1990). Derivation of an angle density function for canopies with ellipsoidal leaf angle distributions. *Agricultural and Forest Meteorology, 49*, 173-176

Chianucci, F., Pisek, J., Raabe, K., Marchino, L., Ferrara, C., & Corona, P. (2018). A dataset of leaf inclination angles for temperate and boreal broadleaf woody species. *Annals of Forest Science, 75*, 50-50

Kuusk, A. (2001). A two-layer canopy reflectance model. *Journal of Quantitative Spectroscopy and Radiative Transfer, 71*, 1-9

Lawrence, D.M., Fisher, R.A., Koven, C.D., Oleson, K.W., Swenson, S.C., Bonan, G., Collier, N., Ghimire, B., Van Kampenhout, L., & Kennedy, D. (2019). The Community Land Model version 5: Description of new features, benchmarking, and impact of forcing uncertainty. *Journal of Advances in Modeling Earth Systems, 11*, 4245-4287

le Maire, G., Marsden, C., Verhoef, W., Ponzoni, F.J., Lo Seen, D., Bégué, A., Stape, J.-L., &

Nouvellon, Y. (2011). Leaf area index estimation with MODIS reflectance time series and model
inversion during full rotations of Eucalyptus plantations. *Remote Sensing of Environment, 115*,
586-599

Li, S., Fang, H., & Zhang, Y. (2023). Determination of the Leaf Inclination Angle (LIA) through Field
and Remote Sensing Methods: Current Status and Future Prospects. *Remote Sensing, 15*, 946

Liu, J., Skidmore, A.K., Wang, T., Zhu, X., Premier, J., Heurich, M., Beudert, B., & Jones, S. (2019).
Variation of leaf angle distribution quantified by terrestrial LiDAR in natural European beech
forest. *ISPRS Journal of Photogrammetry and Remote Sensing, 148*, 208-220

Majasalmi, T., & Bright, R.M. (2019). Evaluation of leaf-level optical properties employed in land
surface models – example with CLM 5.0. *Geoscientific Model Development Discussions*, 1-24

Myneni, R.B., Hoffman, S., Knyazikhin, Y., Privette, J.L., Glassy, J., Tian, Y., Wang, Y., Song, X.,
Zhang, Y., Smith, G.R., Lotsch, A., Friedl, M., Morisette, J.T., Votava, P., Nemani, R.R., &
Running, S.W. (2002). Global products of vegetation leaf area and fraction absorbed PAR from
year one of MODIS data. *Remote Sensing of Environment, 83*, 214-231

Pisek, J., & Adamson, K. (2020). Dataset of leaf inclination angles for 71 different Eucalyptus species.
*Data Brief, 33*, 106391

Pisek, J., Diaz-Pines, E., Matteucci, G., Noe, S., & Rebmann, C. (2022). On the leaf inclination angle
distribution as a plant trait for the most abundant broadleaf tree species in Europe. *Agricultural
and Forest Meteorology, 323*

Raabe, K., Pisek, J., Sonnentag, O., & Annuk, K. (2015). Variations of leaf inclination angle
distribution with height over the growing season and light exposure for eight broadleaf tree
species. *Agricultural and Forest Meteorology, 214-215*, 2-11

Tang, H., Ganguly, S., Zhang, G., Hofton, M.A., Nelson, R.F., & Dubayah, R. (2016). Characterizing
leaf area index (LAI) and vertical foliage profile (VFP) over the United States. *Biogeosciences,
13*, 239-252

Toda, M., Ishihara, M.I., Doi, K., & Hara, T. (2022). Determination of species-specific leaf angle
distribution and plant area index in a cool-temperate mixed forest from UAV and upward-pointing
digital photography. *Agricultural and Forest Meteorology, 325*

Verhoef, W., Jia, L., Xiao, Q., & Su, Z. (2007). Unified Optical-Thermal Four-Stream Radiative
Transfer Theory for Homogeneous Vegetation Canopies. *IEEE Transactions on Geoscience and
Remote Sensing, 45*, 1808-1822

Wang, W.M., Li, Z.L., & Su, H.B. (2007). Comparison of leaf angle distribution functions: Effects on
extinction coefficient and fraction of sunlit foliage. *Agricultural and Forest Meteorology, 143*,
106-122

Wilson, J.W. (1959). Analysis of the spatial distribution of foliage by two-dimensional point quadrats.
*New Phytologist, 58*, 92-99

Yan, K., Pu, J., Park, T., Xu, B., Zeng, Y., Yan, G., Weiss, M., Knyazikhin, Y., & Myneni, R.B. (2021).
Performance stability of the MODIS and VIIRS LAI algorithms inferred from analysis of long
time series of products. *Remote Sensing of Environment, 260*

Yesson, C., Brewer, P.W., Sutton, T., Caithness, N., Pahwa, J.S., Burgess, M., Gray, W.A., White, R.J.,
Jones, A.C., & Bisby, F.A. (2007). How global is the global biodiversity information facility?
*PLoS ONE, 2*, e1124

Zhang, Y., Tang, L., Liu, X., Liu, L., Cao, W., & Zhu, Y. (2017). Modeling the leaf angle dynamics in rice plant. *PLoS ONE, 12*, 1-13

Zhao, J., Li, J., Liu, Q., Xu, B., Yu, W., Lin, S., & Hu, Z. (2020). Estimating fractional vegetation cover from leaf area index and clumping index based on the gap probability theory. *International*

*Journal of Applied Earth Observation and Geoinformation, 90*, 102-112

---

## Author Comment (AC2)

**Referee #2**

This study compiled global Leaf Inclination Angle (LIA) field measurements and produced the first global 500 m LIA dataset using machine learning. The dataset was evaluated with the nadir leaf projection function, comparing it against high-resolution reference data, and the global LIA patterns across different biomes were further analyzed. While the study is intriguing and generally well-written, I have significant concerns regarding the reliability of this static, machine learning-based product, particularly due to the dynamic nature of LIA at the leaf level, limitations in scaling field measurements to the canopy and ecosystem level, and the lack of effective input data at the global scale. My specific concerns are outlined below:

We thank the referee for the insightful comments which help us to further improve the manuscript. We fully understand the referee's concerns and have provided detailed explanations below.

1. **Dynamic Nature of Leaf-Level LIA:** LIA is highly variable within a canopy, even for a single species. Observing a tree canopy, one can easily notice the variation in leaf inclination. To minimize self-shading or optimize light capture, sun and shade leaves on the same plant may have different inclinations. Moreover, LIA can change throughout the day to track the sun's movement, across growing seasons, and with leaf age and developmental stages. Under stress conditions, such as water scarcity or extreme temperatures, plants may adjust their leaf angles to reduce water loss or mitigate heat stress by altering turgor pressure. Additionally, variability in LIA is influenced by branching patterns, stem elongation, and species-specific genetic traits like phototropism and heliotropism. Given this variability, treating LIA as a static structural trait oversimplifies its inherently dynamic nature.

We agree with the referee's comments about the dynamic nature of leaf LIA. For plant physiologists, it is well known that LIA is influenced by environmental conditions and shows temporal variation.

In this study, LIA is the mean leaf inclination angle (MLA) of all leaves at the canopy or pixel scale, not for a single leaf. For a site, the LIA of multiple leaves at different heights and orientations are obtained and averaged to obtain a robust MLA (Chianucci et al. 2018; Pisek and Adamson 2020). The MLA partly mitigates the impact of height, sunlit and shaded leaves, branching patterns, stem elongation, and species-specific genetic traits like phototropism and heliotropism. This kind of mean LIA is desperately wanted in many remote sensing and land surface modeling studies (Lawrence et al. 2019; Li et al. 2023; Majasalmi and Bright 2019; Tang et al. 2016; Zhao et al. 2020). In those studies, LIA is commonly assumed constant (spherical distribution, 57.3 degrees) or biome type-specific (assigning a constant value for each biome). Indeed, these assumptions may not represent the true field measurements (Tables 3 and 4). Our objective is to provide a more realistic global MLA map for remote sensing and land surface modeling studies.

In this study, the LIA seasonal variations were not considered in the global LIA map because of the lack of seasonal LIA measurements. As a matter of fact, temporal LIA variations are usually small, except under extreme situations (unusual). For example, the LIA variations of European beech forest and eucalyptus in different successional stages are less than 10 degrees (le Maire et al. 2011; Liu et al. 2019; Raabe et al. 2015). Crops generally show higher LIA variations than non-crops (Biskup et al. 2007; Zhang et al. 2017). Therefore, many studies have considered LIA as a species-specific static trait when there are no seasonal field measurements (Pisek et al. 2022; Raabe et al. 2015; Toda et al. 2022).

The global LIA map derived in this study is consistent with field measurements (Tables 3 and 4). This is a significant improvement compared to existing static simplifications (Lawrence et al. 2019; Li et al. 2023; Majasalmi and Bright 2019; Tang et al. 2016; Zhao et al. 2020). In a forthcoming study, we plan to retrieve LIA from remote sensing and the temporal LIA variation will be considered.

Thanks to the referee's comment, we have revised the manuscript (Line 151).

> *Many studies have treated LIA as a species-specific static trait and ignored within-species variations when LIA measurements are limited (Pisek et al., 2022; Toda et al., 2022; Raabe et al., 2015). Following the rationale, the spatial coverage of LIA measurements was expanded, and those records without location information were utilized (section 2.1.1).*

2. **Upscaling LIA Field Measurements:** The LIA field measurements from the TRY database seem to be primarily site-specific. The method used to upscale these measurements from the leaf level to the canopy and ecosystem scales is crucial for modeling accuracy, yet it is unclear in this study. The approach of using a weighted average of Enhanced Vegetation Index (EVI) to scale LIA from 30 m to 500 m, as per equation (1), raises concerns. What is the solid physical or physiological rationale for this upscaling method? Without a clear justification, this approach appears problematic.

In field measurement, the entire canopy LIA is calculated as the average of all measured leaf LIAs weighted by leaf area (de Wit 1965; Zou et al. 2014). Leaves with larger areas have higher weights. Upscaling LIA from 30 m to 500 m follows the same rationale as that from leaf to canopy scale. For a 30 m pixel with a higher leaf area index (LAI), the weight of the pixel is higher. Considering that a linear relationship exists between LAI and enhanced vegetation index (EVI2) (Alexandridis et al. 2019; Dong et al. 2019), the LIA was upscaled by EVI2 (Eq. (1)).

Following the suggestion, we have explained in the manuscript (Line 165).

> *In field measurement, the entire canopy LIA is calculated as the average of all measured leaf LIAs weighted by leaf area (Zou et al., 2014; De Wit, 1965). Leaves with larger areas have higher weights. Upscaling LIA from 30 m to 500 m follows the same rationale as that from leaf to canopy scale. For a 30 m pixel with a higher LAI, the weight of the pixel is higher. Therefore, the 500 m MLA was computed as the weighted average of the enhanced vegetation index (EVI2) considering a linear relationship between LAI and EVI2 (Dong et al., 2019; Alexandridis et al., 2019).*

3. **Coarse Resolution and Low-Signal Inputs in the Model:** LIA provides detailed structural information at the leaf level. When using a machine learning model, how did the authors ensure that the global model inputs listed in Table 1 accurately represent such low-signal information (also the variations mentioned in comment #1) at a coarse spatial resolution, which is significantly larger than the leaf level? Importantly, the MODIS LAI product does not reliably capture LIA in its algorithm. Furthermore, as seen in Figure 6, NDVI and precipitation are identified as major factors controlling LIA. What is the specific basis for this, given that both factors exhibit strong seasonal dynamics? Overall, I think that current optical remote sensing systems, such as MODIS and Landsat, lack the capability to capture the subtle structural signal of LIA, as they were not designed for this purpose.

We agree with the referee that MODIS and Landsat are not designed for estimating LIA.

In this study, the MODIS LAI was only used for the upscaling evaluation of G(0) (Line 219). In the MODIS LAI algorithm, a biome-specific static LIA was used as a priori (Myneni et al. 2002). This biome-specific LIA is very rough and should (and can) be improved. It is our goal to generate global pixel-scale LIA.

The correlation between LIA and NDVI or precipitation has been reported in many simulation and field studies (Dong et al. 2019; Jacquemoud et al. 1994; Liu et al. 2012; Zou and Mõttus 2015). This has been explained in section 4.2. Higher LIA means lower radiation interception, more NIR downward radiation, and lower NIR reflectance (Liu et al. 2012). This results in negative correlations between MLA and NIR reflectance and vegetation index. The negative correlation between MLA and precipitation relates to vegetation adaptation. Under suitable climate conditions, horizontal leaves can make better usage of precipitation and increase the photosynthesis rate (King 1997; van Zanten et al. 2010). Therefore, in this study, the mean and stand deviation of NDVI and precipitation time series were selected to predict LIA. The mean NDVI and precipitation represent the average situation for a specific area and correspond to the typical global LIA.

In canopy radiation transfer, canopy structure parameters, including leaf area index, LIA, and clumping index jointly determine the canopy reflectance (Liang 2005; Ross 1981; Verhoef 1984). Previous studies have shown that multi-angle reflectance is sensitive to LIA and can be used to derive the latter (Goel and Thompson 1984; Jacquemoud et al. 1994; Jacquemoud et al. 2009; Li et al. 2023). Since MODIS has multiangle observations, the multiangle information provided in the BRDF product (MCD43A1 C6.1) was used here as LIA predictors in this study. In contrast, Landsat lacks a multiangle view and was rarely used for LIA estimation.

**Reference**

Alexandridis, T.K., Ovakoglou, G., & Clevers, J.G.P.W. (2019). Relationship between MODIS EVI and LAI across time and space. *Geocarto International, 35*, 1385-1399

Biskup, B., Scharr, H., Schurr, U., & Rascher, U. (2007). A stereo imaging system for measuring structural parameters of plant canopies. *Plant, Cell and Environment, 30*, 1299-1308

Chianucci, F., Pisek, J., Raabe, K., Marchino, L., Ferrara, C., & Corona, P. (2018). A dataset of leaf inclination angles for temperate and boreal broadleaf woody species. *Annals of Forest Science, 75*, 50-50

de Wit, C.T. (1965). Photosynthesis of leaf canopies. In: Pudoc

Dong, T., Liu, J., Shang, J., Qian, B., Ma, B., Kovacs, J.M., Walters, D., Jiao, X., Geng, X., & Shi, Y. (2019). Assessment of red-edge vegetation indices for crop leaf area index estimation. *Remote Sensing of Environment, 222*, 133-143

Goel, N.S., & Thompson, R.L. (1984). Inversion of vegetation canopy reflectance models for
estimating agronomic variables. V. Estimation of leaf area index and average leaf angle using
measured canopy reflectances. *Remote Sensing of Environment, 16*, 69-85

Jacquemoud, S., Flasse, S., Verdebout, J., & Schmuck, G. (1994). Comparison of Several Optimization
Methods To Extract Canopy Biophysical Parameters - Application To Caesar Data, 291-298

Jacquemoud, S., Verhoef, W., Baret, F., Bacour, C., Zarco-Tejada, P.J., Asner, G.P., François, C., &
Ustin, S.L. (2009). PROSPECT+SAIL models: A review of use for vegetation characterization.
*Remote Sensing of Environment, 113*, S56-S66

King, D.A. (1997). The Functional Significance of Leaf Angle in Eucalyptus. *Australian Journal of*
*Botany, 45*, 619-639

Lawrence, D.M., Fisher, R.A., Koven, C.D., Oleson, K.W., Swenson, S.C., Bonan, G., Collier, N.,
Ghimire, B., Van Kampenhout, L., & Kennedy, D. (2019). The Community Land Model version 5:
Description of new features, benchmarking, and impact of forcing uncertainty. *Journal of*
*Advances in Modeling Earth Systems, 11*, 4245-4287

le Maire, G., Marsden, C., Verhoef, W., Ponzoni, F.J., Lo Seen, D., Bégué, A., Stape, J.-L., &
Nouvellon, Y. (2011). Leaf area index estimation with MODIS reflectance time series and model
inversion during full rotations of Eucalyptus plantations. *Remote Sensing of Environment, 115*,
586-599

Li, S., Fang, H., & Zhang, Y. (2023). Determination of the Leaf Inclination Angle (LIA) through Field
and Remote Sensing Methods: Current Status and Future Prospects. *Remote Sensing, 15*, 946

Liang, S. (2005). *Quantitative remote sensing of land surfaces*. John Wiley & Sons

Liu, J., Pattey, E., & Jégo, G. (2012). Assessment of vegetation indices for regional crop green LAI
estimation from Landsat images over multiple growing seasons. *Remote Sensing of Environment,*
*123*, 347-358

Liu, J., Skidmore, A.K., Wang, T., Zhu, X., Premier, J., Heurich, M., Beudert, B., & Jones, S. (2019).
Variation of leaf angle distribution quantified by terrestrial LiDAR in natural European beech
forest. *ISPRS Journal of Photogrammetry and Remote Sensing, 148*, 208-220

Majasalmi, T., & Bright, R.M. (2019). Evaluation of leaf-level optical properties employed in land
surface models – example with CLM 5.0. *Geoscientific Model Development Discussions*, 1-24

Myneni, R.B., Hoffman, S., Knyazikhin, Y., Privette, J.L., Glassy, J., Tian, Y., Wang, Y., Song, X.,
Zhang, Y., Smith, G.R., Lotsch, A., Friedl, M., Morisette, J.T., Votava, P., Nemani, R.R., &
Running, S.W. (2002). Global products of vegetation leaf area and fraction absorbed PAR from
year one of MODIS data. *Remote Sensing of Environment, 83*, 214-231

Pisek, J., & Adamson, K. (2020). Dataset of leaf inclination angles for 71 different Eucalyptus species.
*Data Brief, 33*, 106391

Pisek, J., Diaz-Pines, E., Matteucci, G., Noe, S., & Rebmann, C. (2022). On the leaf inclination angle
distribution as a plant trait for the most abundant broadleaf tree species in Europe. *Agricultural*
*and Forest Meteorology, 323*

Raabe, K., Pisek, J., Sonnentag, O., & Annuk, K. (2015). Variations of leaf inclination angle
distribution with height over the growing season and light exposure for eight broadleaf tree
species. *Agricultural and Forest Meteorology, 214-215*, 2-11

Ross, J. (1981). *The radiation regime and architecture of plant stands*. Springer Science & Business

Media

Tang, H., Ganguly, S., Zhang, G., Hofton, M.A., Nelson, R.F., & Dubayah, R. (2016). Characterizing
  leaf area index (LAI) and vertical foliage profile (VFP) over the United States. *Biogeosciences,*
  *13*, 239-252

Toda, M., Ishihara, M.I., Doi, K., & Hara, T. (2022). Determination of species-specific leaf angle
  distribution and plant area index in a cool-temperate mixed forest from UAV and upward-pointing
  digital photography. *Agricultural and Forest Meteorology, 325*

van Zanten, M., Pons, T.L., Janssen, J.A.M., Voesenek, L.A.C.J., & Peeters, A.J.M. (2010). On the
  Relevance and Control of Leaf Angle. *Critical Reviews in Plant Sciences, 29*, 300-316

Verhoef, W. (1984). Light scattering by leaf layers with application to canopy reflectance modeling:
  The SAIL model. *Remote Sensing of Environment, 16*, 125-141

Zhang, Y., Tang, L., Liu, X., Liu, L., Cao, W., & Zhu, Y. (2017). Modeling the leaf angle dynamics in
  rice plant. *PLoS ONE, 12*, 1-13

Zhao, J., Li, J., Liu, Q., Xu, B., Yu, W., Lin, S., & Hu, Z. (2020). Estimating fractional vegetation cover
  from leaf area index and clumping index based on the gap probability theory. *International*
  *Journal of Applied Earth Observation and Geoinformation, 90*, 102-112

Zou, X., & Mõttus, M. (2015). Retrieving crop leaf tilt angle from imaging spectroscopy data.
  *Agricultural and Forest Meteorology, 205*, 73-82

Zou, X., Mõttus, M., Tammeorg, P., Torres, C.L., Takala, T., Pisek, J., Mäkelä, P., Stoddard, F.L., &
  Pellikka, P. (2014). Photographic measurement of leaf angles in field crops. *Agricultural and*
  *Forest Meteorology, 184*, 137-146

---

## Referee Report (RR1)

After reviewing the authors' responses, I find that two of my original comments have been adequately addressed. However, one critical concern regarding the upscaling approach remains insufficiently addressed, and the resultant LIA at the ecosystem or grid scale is still rather confusing. Additionally, the authors' major responses are not clearly reflected or integrated into the revised manuscript. Below are my specific comments:

1). Upscaling LIA from the leaf level to the canopy or larger ecosystem scales is inherently challenging. Although the authors provide some clarification, their initial upscaling step remains overly simplistic, making it difficult to grasp what the "ecosystem-level LIA" truly represents. Traditionally, LIA at the canopy scale can be defined as the average LIA of each leaf (Eq. 1). However, because counting individual leaves (N) is often impractical, the authors employ a leaf-area-weighted approach for MLA. If I understand right, this equation can be defined by Eqs. 2 & 3.

$$MLA = \frac{\sum_i LIAi}{N} \qquad (1)$$

$$MLA = \frac{\sum_j LIAi*LAj}{N*LAmean} = \frac{\sum_j LIAi*LAj}{LAI*Canopy\_size} \qquad (2)$$

$$LAI = N*LAmean/canopy\_size = EVI2*a + b \qquad (3)$$

Where MLA is mean inclination angle, j is the jth leaf, LIA is leaf inclination angle, N is number of leaves within a canopy, LA is single leaf area, LAI is the ecosystem-level standard leaf area index (m2/m2), canopy_size is the projected area onto the ground for a specific canopy; a and b are the linear coefficients between EVI2 and LAI (if the linear relationship holds true).

Eqs. (2) and (3) theoretically support the upscaling of LIA from the leaf to the canopy level, and by extension from the canopy to 30 m and from 30 m to 500 m. However, the authors used a simplified form of Eq (1) in the manuscript to upscale from 30m to 500m. It is hard to persuade me this equation is equivalent to the Eqs (2-3) mentioned above, especially given the existence of the interception of b and missing variable of leaf number.

In addition, the authors did not mention the details of upscaling from the canopy to 30m. As a result, the MLA on the 500m derived here and further used to training the model is difficult to interpret, which is apparently different from the LIA at the leaf level. I encourage the authors to more rigorously evaluate their upscaling methodology, discussing the assumptions and uncertainties introduced at each scale and from different data sources.

2). The authors argued that "higher LIA means lower radiation interception, more NIR downward radiation, and lower NIR reflectance", thus negatively correlated with NDVI. However, a higher LIA could also reduce red reflectance, potentially complicating how

NDVI encapsulates leaf angle information. Moreover, as NDVI is designed as a normalized index, one might expect it to diminish the effects of incidence angles in BRDF data (MCD43A1). Considering the global availability of GEDI lidar (with a 25 m footprint) and its known sensitivity to canopy structure (e.g., height), it would be worthwhile to test whether GEDI can provide stronger signals of LIA than optical-only approaches. Such an investigation could bolster the validation or derivation of the first global MLA map.

3). In Table 1, MCD12Q1 and MCD43A4 are listed as Collection 6, while other MODIS products are Collection 6.1. The discrepancy in MODIS versions needs clarification. Furthermore, MODIS BRDF (MCD43) and surface reflectance products can be contaminated by clouds, especially in tropical regions. The manuscript should explicitly describe how these cloud gaps or low-quality observations were handled to ensure their usage in the subsequent modeling.

4). As the first global MLA product, it would be valuable to include an uncertainty assessment layer. This might account for the uncertainties stemming from (1) the upscaling approach, (2) the machine learning model, and (3) data inputs. Presenting an explicit uncertainty layer would markedly improve the credibility and potential applications of this novel dataset.

---

## Author Response (AR2)

- 1 Topic editor
- 2

The manuscript presents a novel effort to generate the first global 500m resolution mean leaf inclination angle (MLA) product, along with associated leaf projection function data. This work contributes significantly to the field by addressing gaps in phenological and vegetation structure parameters critical for land surface models and remote sensing applications. Both reviewers acknowledge the scientific novelty of the study, the rigorous methodological approach, and its potential to improve vegetation modeling and remote sensing parameter inversion.

However, the reviewers raised significant concerns about methodological robustness, 12 particularly regarding upscaling LIA field measurements, handling coarse-resolution 13 data, and the selection of predictive features. Reviewer 1 emphasized the need for better 14 clarification and testing of the upscaling processes and questioned the reliance on 15 MODIS-based data for capturing LIA signals. Reviewer 2 highlighted the importance of incorporating additional remote sensing parameters and addressing uncertainties in 16 17 the data sources. Furthermore, both reviewers pointed out the need to better validate 18 the product and address apparent biases, such as overestimation in specific cases.

While the study provides a strong foundation, addressing these concerns through more rigorous uncertainty analysis, methodological refinement, and clearer discussion of data limitations will be necessary for the next revision. The potential to refine global vegetation models and ecological understanding underscores the importance of this work, warranting reconsideration after major revisions.

We thank the topic editor for the recognition and professional processing. We fully understand the concerns raised by the reviewers and have carefully addressed these issues in this revision round.

Some major revisions were made in the revised version:

- 31 (1) The concerns regarding the upscaling approach and modeling inputs have been32 explained.
- 33 (2) The uncertainties in the data sources and upscaling process have been analyzed34 (section 4.4).
- 35 (3) The necessity of introducing additional remote sensing parameters to MLA36 mapping has been analyzed.

- 37 (4) The validity of using G(0) validation to evaluate MLA indirectly has been
demonstrated (section 4.1).
- 39 (5) A regional analysis of variable importance has been conducted (section 4.2).
- 40 (6) Some other revisions for the manuscript and supplement material have been made.
- 41
- 42 Please see the itemized responses below.
- 43

- 44 Anonymous Referee #2
- 45

I agree that Leaf Inclination Angle (LIA) is indeed a critical parameter for global land surface models, such as Dynamic Global Vegetation Models (DGVM). However, after reviewing the authors' responses, I find that most of my original comments remain unaddressed or insufficiently tested. Although this is the first global LIA product, as the authors claim, potential issues in both the upscaling approach and modeling inputs raise substantial concerns about its reliability.

We thank the referee for the recognition and thorough comments that helped us improve the manuscript. We fully understand the referee's concerns regarding upscaling approach and modeling inputs and have provided detailed explanations below. We think much of the misunderstanding is caused by the different requirements for canopy traits between the remote sensing and plant physiology communities.

Regarding my second comment on "Upscaling LIA Field Measurements," the authors 60 mentioned that "in field measurements, the entire canopy LIA is calculated as the average of all measured leaf LIAs weighted by leaf area." I question why leaf area, 61 62 rather than leaf number, is used for this weighting. Given the highly variable nature of 63 LIA within a canopy and across species and ecosystems (as noted in my first comment), 64 upscaling LIA measurements from site level to 30m, and subsequently to 500m scales, 65 is a crucial initial step. Yet, it remains unclear how the authors executed these steps or assessed the associated uncertainties. Using leaf area rather than leaf number for 66 67 weighting raises concerns about the representativeness of the measurements.

This is an excellent point. In this study, two different weighting methods were used: (1) 70 from leaf to canopy scale, leaf number was used; and (2) from 30 m to 500 m, leaf area 71 was used. From leaf to canopy scale, the entire canopy LIA is commonly calculated as 72 the average of all measured leaf LIAs weighted by leaf area in the remote sensing 73 community (Zou et al., 2014; De Wit, 1965; Yan et al., 2021). For example, Yan et al. 74 (2021) stated that the final leaf angle distribution is obtained by weighting the relative areas with different leaf inclination angles. Leaves with larger areas contribute more to 75 76 photosynthesis and have higher weights. In practice, because of the difficulty in leaf 77 area measurement, especially for a large number of leaves, the variability of leaf areas 78 within a canopy is often ignored and the areas of all leaves are assumed similar. In this 79 case, the canopy LIA can be simplified as the average LIA weighted by leaf number 80 (Ryu et al., 2010; Pisek et al., 2011; Chianucci et al., 2018). Therefore, in this study, the canopy LIA measurements were also obtained by weighting leaf LIA with leafnumber.

- 83
- 84 The obtained canopy LIA measurement was used to represent the LIA at the 30 m pixel
- 85 level considering the representativeness. The LIA upscaling from 30 m to 500 m was
- 86 weighted by the 30 m leaf area index (using EVI2 as a proxy). Leaf area index (LAI) is
- 87 defined as the half of green leaf area on the unit of ground area and is similar to leaf
- 88 number/density to some extent (Fang et al., 2019). High leaf number typically means
- 89 high LAI. For a 30 m pixel with a higher LAI, its weight/contribution to the 500 m scale
- 90 LIA is also higher (Fig. R1).
- 91

Fig. R1. Schematic of LIA upscaling from 30 m to 500 m. The green and yellow colorsdenote high and low leaf area index, respectively.

When one plant function type (PFT) within a 500 m pixel has no LIA measurement, the
LIA of the PFT was assigned with the value of its nearest neighbor within 100 km with
the same PFT. This upscaling practice has been used to map global leaf traits (specific
leaf area, leaf dry matter content, leaf nitrogen and phosphorus content per dry mass,
and leaf nitrogen/phosphorus ratio) at 500 m spatial resolution (Moreno-Martínez et al.,
2018).

For my third comment on "Coarse Resolution and Low-Signal Inputs in the Model," I
feel the authors' response is also lacking. The BRDF product primarily normalizes
surface reflectance by mitigating inconsistencies arising from varying sun and sensor angles. With the current 500m resolution, the fine-scale signal of LIA is vulnerable to 107 interference from surface structures, such as canopy heterogeneity, surface roughness, 108 height, clustering, branch structures, and terrain. I am skeptical that MODIS's passive optical sensor can capture LIA signals effectively (Such signals may be better detected 109 by radar or lidar data). Additionally, the claim that "Under suitable climate conditions, 110 111 horizontal leaves can make better usage of precipitation and increase the photosynthesis 112 rate" is problematic. Water use efficiency is unlikely to be closely related to leaf angles. 113 Currently, NDVI is tested as the primary indicator for LIA, yet NDVI primarily reflects 114 chlorophyll content, which is largely decoupled from information on leaf inclination 115 angle.

Our study has used the BRDF model parameters product (MCD43A1 C6.1, the variables. 118 https://lpdaac.usgs.gov/products/mcd43a1v006/) as predictive MCD43A1 provides three model weighting parameters for different kernels (isotropic, 119 120 volumetric, and geometric), which can be employed to compute the directional 121 reflectance (Schaaf et al., 2002). We suspect that the referee has mistaken the BRDF 122 product as the Nadir Bidirectional Reflectance Distribution Function (BRDF)-Adjusted Reflectance (NBAR) (MCD43A4, https://lpdaac.usgs.gov/products/mcd43a4v006/), 123 124 which is derived from MCD43A1 but is normalized to a unified nadir viewing geometry. 125 It is true that the nadir reflectance is difficult to retrieve LIA, as demonstrated by a previous study (Bayat et al., 2018). Nonetheless, the directional reflectance variation is 126 127 sensitive to LIA (Fig. R2) and has been used to derive LIA from many passive optical sensors (Jacquemoud et al., 2009; Goel and Thompson, 1984; Jacquemoud et al., 1994; 128 129 Li et al., 2023).

Fig. R2. Contribution of LIA (%) to the top-of-canopy directional reflectance
(excerpted from Jacquemoud et al. (2009)). The solar zenith angle (31.6°) is indicated
by a star.

At 500 m, the multi-angle reflectance information is related to the average canopy LIA 136 137 at the same scale. The terrain variables were introduced in the LIA prediction, which 138 partly mitigates the interference from surface structures. As illustrated in Figs. 6 and 7, 139 the BRDF parameters at 500 m scale are sensitive to LIA, further indicating the validity 140 of this practice. In addition, detecting LIA with radar remains in the simulation stage 141 (Lang and Saleh, 1985) and no practical studies have been reported. Point cloud LiDAR 142 has been used to measure LIA accurately but is limited to a local scale due to the 143 limitation of the sensor platform (Zheng and Moskal, 2012; Bailey and Mahaffee, 2017; 144 Itakura and Hosoi, 2019). Currently, no study has used spaceborne LiDAR to estimate LIA.

- 145
- 146

We agree that the original claim "Under suitable climate conditions, horizontal leaves 148 can make better usage of precipitation and increase the photosynthesis rate" is not solid. 149 Under suitable climate conditions (radiation, precipitation, and temperature), the 150 elements required for photosynthesis are satisfied, and horizontal leaves are formed to 151 absorb more radiation and increase the photosynthesis rate (Van Zanten et al., 2010; 152 King, 1997). We have rephrased it as (line 379)

- 153 "Under suitable climate conditions (radiation, precipitation, and temperature), 154 horizontal leaves are formed to absorb more radiation and increase the 155 photosynthesis rate (Van Zanten et al., 2010; King, 1997)".
- 156

We agree NDVI is related to chlorophyll content, but only when LAI is high (LAI >= 157 158 4) (Fig. R3). When LAI < 4, NDVI is strongly coupled with LIA. Globally, the global 159 mean LAI is ~1.20 and high LAI (>=4) only occupies a tiny fraction (Fang et al., 2021). NDVI is frequently used to retrieve canopy structural parameters, such as leaf area 160 161 index, and fractional vegetation cover (Carlson and Ripley, 1997; Carlson et al., 1994; Wang et al., 2005), but was rarely used for the chlorophyll content, which is more 162 163 closely related to various chlorophyll indexes formulated by green, red, NIR, and red-164 edge bands (Dong et al., 2019; Haboudane et al., 2002; Gitelson et al., 2003; Wu et al., 165 2008). In this study, NDVI is an important contributor to the LIA prediction (Figs. 6 and 7). The correlation between LIA and NDVI has been reported in many simulation 166 167 and field studies (Fig. R3) (Zou and Mõttus, 2015; Liu et al., 2012; Dong et al., 2019; 168 Jacquemoud et al., 1994) and has been explained in section 4.2. Higher LIA means lower radiation interception, more NIR downward radiation, and lower NIR reflectance 169 170 (Liu et al., 2012). This results in a negative correlation between LIA and NDVI. In addition, besides NDVI, we have used many other important indicators (includingclimate, BRDF, terrain) that are related to LIA to predict MLA.

---

## Author Response (AR3)

**Topic editor**

Public justification (visible to the public if the article is accepted and published):

The manuscript presents a novel approach to estimating global Mean Leaf Inclination Angle (MLA) using satellite-derived vegetation indices and machine learning. Both reviewers acknowledge the improvements made in response to their initial comments, with many concerns adequately addressed. However, several key issues remain unresolved, warranting further revision. Reviewer 1 highlights the need for a clearer justification of the choice of EVI over other vegetation indices such as NDVI, particularly in light of recent research on vegetation index error propagation and saturation effects. Additionally, a more detailed explanation of the nonlinear LAI-EVI relationship and its saturation phenomenon is necessary. Reviewer 2 raises significant concerns regarding the upscaling methodology, particularly the transition from leaf-level LIA to ecosystem-scale MLA, emphasizing the need for a more rigorous discussion of assumptions and uncertainties. Furthermore, greater integration of responses into the manuscript, clarification of MODIS product versions, and a dedicated uncertainty assessment layer would strengthen the study's credibility. Given these remaining concerns, another major revision is necessary to ensure the robustness and transparency of the methodology, as well as to enhance the interpretability and applicability of the global MLA dataset.

We thank the topic editor for the recognition and professional processing. We fully understand the concerns raised by the reviewers and have carefully addressed these issues in this revision round.

Some major revisions were made in the revised version:
(1) The reasons for the choice of EVI2 and explanations of the nonlinear LAI-EVI2 relationship have been further elaborated in the main text.
(2) The full process of upscaling methodology has been reorganized rigorously to enhance clarity and its assumptions and uncertainty have been discussed.
(3) The uncertainty assessment layers have been added from the perspectives of inputs and the prediction model.
(4) The comments regarding MODIS products, NDVI, and GEDI LiDAR have been addressed.
(5) The responses to reviewers have been greatly integrated into the manuscript.
(6) The data DOI has been updated because of the data upgrade.

After reviewing the authors' responses, I find that two of my original comments have been adequately addressed. However, one critical concern regarding the upscaling approach remains insufficiently addressed, and the resultant LIA at the ecosystem or grid scale is still rather confusing. Additionally, the authors' major responses are not clearly reflected or integrated into the revised manuscript. Below are my specific comments:

We thank the referee for the insightful comments which significantly improved the manuscript. We fully understand the referee's concerns and have provided detailed explanations and revisions below. In addition, the previous major responses to your comments regarding *Upscaling LIA Field Measurements* and *Coarse Resolution and Low-Signal Inputs in the Model* have been integrated into the revised manuscript (Sections 2.2.1, 2.3.1, and 2.3.2).

1). Upscaling LIA from the leaf level to the canopy or larger ecosystem scales is inherently challenging. Although the authors provide some clarification, their initial upscaling step remains overly simplistic, making it difficult to grasp what the "ecosystem-level LIA" truly represents. Traditionally, LIA at the canopy scale can be defined as the average LIA of each leaf (Eq. 1). However, because counting individual leaves (N) is often impractical, the authors employ a leaf-area-weighted approach for MLA. If I understand right, this equation can be defined by Eqs. 2 & 3.

$$MLA = \frac{\sum_i LIA_i}{N} \qquad (1)$$

$$MLA = \frac{\sum_j LIA_j * LA_j}{N * LA_{mean}} = \frac{\sum_j LIA_j * LA_j}{LAI * canopy\_size} \qquad (2)$$

$$LAI = N * LA_{mean}/canopy\_size = EVI2 * a + b \qquad (3)$$

Where MLA is mean inclination angle, j is the jth leaf, LIA is leaf inclination angle, N is number of leaves within a canopy, LA is single leaf area, LAI is the ecosystem-level standard leaf area index (m2/m2), canopy_size is the projected area onto the ground for a specific canopy; a and b are the linear coefficients between EVI2 and LAI (if the linear relationship holds true).

Eqs. (2) and (3) theoretically support the upscaling of LIA from the leaf to the canopy
level, and by extension from the canopy to 30 m and from 30 m to 500 m. However,
the authors used a simplified form of Eq (1) in the manuscript to upscale from 30m to
500m. It is hard to persuade me this equation is equivalent to the Eqs (2-3) mentioned
above, especially given the existence of the interception of b and missing variable of
leaf number.

In addition, the authors did not mention the details of upscaling from the canopy to 30m.
As a result, the MLA on the 500m derived here and further used to training the model
is difficult to interpret, which is apparently different from the LIA at the leaf level. I
encourage the authors to more rigorously evaluate their upscaling methodology,
discussing the assumptions and uncertainties introduced at each scale and from different
data sources.

Thank the referee for this thorough comment. We have reorganized the upscaling
process rigorously to enhance clarity.

From leaf to canopy scale, the entire canopy MLA is commonly calculated as the
average of all measured leaf LIAs weighted by leaf area in the remote sensing
community (Eq. R1) (Zou et al., 2014; De Wit, 1965; Yan et al., 2021). In practice,
because of the difficulty in leaf area measurement, especially for a large number of
leaves, the variability of leaf areas within a canopy is often ignored and the areas of all
leaves are assumed similar. In this case, the canopy LIA can be simplified as the average
LIA weighted by leaf number (Eq. R1) (Ryu et al., 2010; Pisek et al., 2011; Chianucci
et al., 2018):

$$MLA_{canopy} = \frac{\sum_i LIA_i * LA_i}{\sum_i LA_i} = \frac{LA_{mean} * \sum_i LIA_i}{LA_{mean} * N} = \frac{\sum_i LIA_i}{N} \qquad (R1)$$

where $MLA_{canopy}$ is the MLA at canopy scale, $i$ is the $i$th leaf, LIA is leaf inclination
angle, LA is single leaf area, $LA_{mean}$ is the mean leaf area by ignoring the variation
of leaf area within a canopy, N is number of leaves within a canopy.

From the canopy to 30 m scale, the canopy level MLA is regarded as equal to 30 m-
MLA because for MLA measurements, the dominant species was artificially identified
by investigators, and the spatial representativeness at the extent of 30 m is ensured. This
practice has been used in previous studies to derive global maps for various leaf traits (specific leaf area, leaf dry matter content, leaf nitrogen and phosphorus content per dry
mass, and leaf nitrogen/phosphorus ratio) from TRY leaf trait measurements, remote
sensing, and climate data (Moreno-Martínez et al., 2018).

From 30 m to 500 m, the 500 m MLA was formulated as the weighted average of 30 m
MLA by the leaf area of the 30 m pixel (Eq. R2), the same as that from the leaf to
canopy scale. The leaf area of a 30 m pixel can be deduced from the product of leaf area
index (LAI) and the ground area (not the projected area onto the ground for a specific
canopy) of a 30 m pixel according to the definition of LAI (the half of green leaf area
on the unit of ground area) (Eq. R2) (Fang et al., 2019).

$$MLA_{500} = \frac{\sum_j MLA_{30\_j} * LA_{30\_j}}{\sum_j LA_{30\_j}} = \frac{\sum_j MLA_{30\_j} * LAI_{30\_j} * S}{\sum_j LAI_{30\_j} * S} = \frac{\sum_j MLA_{30\_j} * LAI_{30\_j}}{\sum_j LAI_{30\_j}} \qquad (R2)$$

Where $MLA_{500}$ and $MLA_{30}$ represent MLA at 500 m and 30 m scales, $j$ is the $j$th 30
m pixel, $LA_{30\_j}$ is the total leaf area of a 30 m pixel, $LAI_{30\_j}$ is leaf area index
(m2/m2) of a 30 m pixel, $S$ is the ground area of a 30 m pixel.

Assuming LAI=a*EVI2+b and b ≈ 0 (as illustrated in Fig. R1), the MLA at 500 m scale
can be calculated as

$$MLA_{500} = \frac{\sum_j MLA_{30\_j} * EVI2_{30\_j}}{\sum_j EVI2_{30\_j}} \qquad (R3)$$

The linear relationship between LAI and EVI2 is an important assumption in the MLA
upscaling. We have attempted to use the real MODIS LAI-EVI2 relationship (Fig. R1)
from global statistics to correct the MLA upscaling procedure. 2,000 points for each
biome type were randomly sampled and the LAI-EVI2 pairs with good quality per 8
days for these points were extracted. The LAI-EVI2 relationship is nearly linear and
the intercept is close to 0 (Fig. R1).

[Figure]

Fig. R1. The nonlinear relationship between MODIS LAI and EVI2.

Subsequently, we have updated the MLA training samples with the fitted nonlinear relationship (Fig. R1, Eq. R2) and compared the samples to the original samples based on the linear assumption (Eq. R3). The updated samples show high consistency with the original samples (Fig. R2). This may be related to the rigorous sample screening to keep the homogeneity of a 500 m sample, which reduces the impact of the LAI-EVI2

nonlinear relationship by limiting LAI variations within the 500 m pixel. Therefore, the

LAI-EVI2 linear assumption is reasonable.

[Figure]

Fig. R2. The comparison between the updated samples using the LAI-EVI2 relationship and original MLA samples using EVI2. The black dashed and red solid lines represent

1:1 and fitted lines.

In addition, we agree that uncertainty may arise due to the different data sources (from TRY, literature, and manual extraction). We think the predicted MLA is robust to these differences because part of the samples and features are randomly selected in the training process and the random forest algorithm ensembles the predictions from multiple decision trees (Svetnik et al., 2003). We have manually inspected all field LIA data and are confident in their data quality.

Following the comments, we have added a detailed description regarding LIA upscaling in Appendix A and have discussed the uncertainty of the LAI-EVI2 linear relationship assumption in Section 4.4. The uncertainty raised by different data sources has been discussed in Section 4.4.

*Section 4.4 LAI-EVI2 linear relationship assumption*

*We assumed a linear LAI-EVI2 relationship (LAI = a\*EVI2) to upscale MLA from the canopy to 500 m scale (Section 2.3.1 and Appendix A). Global analysis of MODIS LAI and EVI2 products shows a slight non-linear relationship between them (Fig. S8). The non-linear relationship was also used to upscale MLA (Eq. A2) in a side experiment, where the derived MLA was found consistent with the original one (Fig. S9) because of the homogeneity of the 500 m pixel after rigorous sample screening (section 2.3.1). This demonstrates the suitability of the linear assumption.*

*Section 4.4 Different Data Sources*

*Second, three different sources of LIA measurements were gathered with different measurement schemes, and uncertainty may arise because of these differences. The random forest algorithm is robust to these differences because part of the samples and features were randomly selected and the algorithm ensembled the predictions from multiple decision trees (Svetnik et al., 2003). We manually inspected all field LIA data and are confident in their data quality.*

2). The authors argued that "higher LIA means lower radiation interception, more NIR downward radiation, and lower NIR reflectance", thus negatively correlated with NDVI. However, a higher LIA could also reduce red reflectance, potentially complicating how NDVI encapsulates leaf angle information. Moreover, as NDVI is designed as a normalized index, one might expect it to diminish the effects of incidence angles in BRDF data (MCD43A1). Considering the global availability of GEDI lidar (with a 25 m footprint) and its known sensitivity to canopy structure (e.g., height), it would be worthwhile to test whether GEDI can provide stronger signals of LIA than optical-only approaches. Such an investigation could bolster the validation or derivation of the first global MLA map.

We thank the referee for these comments. High LIA results in low NIR reflectance because more NIR downward radiation reaches the soil background and the NIR

reflectance of soil is lower than that of vegetation (Fig. R3). In terms of red reflectance, high LIA means more red radiation penetrates the canopy and the red reflectance of soil is higher than that of vegetation because of the strong leaf absorption in this wavelength (Fig. R3), causing high red reflectance. Therefore, high LIA causes low NDVI

according to its definition ((NIR-Red)/(NIR+Red)). We have rephrased the original sentence in Section 4.2:

*Higher MLA means lower radiation interception, more NIR and red downward*

*radiations reach the soil background. This causes lower NIR and higher red*

*reflectance because the soil background typically has lower (higher) reflectance*

*for NIR (red) (Siegmund and Menz, 2005). This results in negative correlations*

*between MLA and NIR reflectance and NDVI (Liu et al., 2012).*

[Figure]

Fig. R3 The typical spectral reflectance curves of soil, vegetation, and water. (adapted from (Siegmund and Menz, 2005).

This study used the nadir reflectance product (MCD43A4) corresponding to local solar noon to calculate NDVI; therefore, the solar-viewing geometry of NDVI is consistent.

The consistent geometry and the normalization characteristic of NDVI diminish the angular variation but ensure consistency. In addition, NDVI negatively correlates to

LIA as stated above, and contains vegetation type and vegetation cover information,
which was combined with BRDF and other features to improve MLA mapping.
GEDI LiDAR is indeed a powerful sensor to detect canopy structures, such as tree
height, fractional vegetation cover, and LAI profile (Tang et al., 2016; Dubayah et al.,
2020). Estimating MLA from GEDI LiDAR is an interesting and challenging topic, and
no related studies have been reported due to the difficulty in decoupling MLA from LAI
by the GEDI LiDAR waveform data. In the GEDI LAI retrieval algorithm, MLA is a
key input and is assumed as constant (57.3°) due to the lack of MLA information (Tang
et al., 2016). The MLA map generated in this study can be used to improve this issue.
3). In Table 1, MCD12Q1 and MCD43A4 are listed as Collection 6, while other MODIS
products are Collection 6.1. The discrepancy in MODIS versions needs clarification.
Furthermore, MODIS BRDF (MCD43) and surface reflectance products can be
contaminated by clouds, especially in tropical regions. The manuscript should explicitly
describe how these cloud gaps or low-quality observations were handled to ensure their
usage in the subsequent modeling.
The MCD12Q1 C6 and MCD43A4 V6 were employed in this study (Table 1) because
the Collection 6.1 versions were unavailable on the Google Earth Engine when
conducting the MLA mapping. The official document indicates that only minor
reprocessing including calibration change and polarization correction was adopted in
the upgrading from Collection 6 to 6.1, while the MCD12Q1 and MCD43A4 algorithms
remain                                                                    unchanged
(https://landweb.modaps.eosdis.nasa.gov/data/userguide/MODIS_Land_C61_Change
s.pdf). Previous validation studies with ground truth references have demonstrated that
the improvement from C6 to C6.1 (aerosol products, land surface temperature products)
is very small ($\triangle R^2 < 0.02$), and the accuracy may even decrease (Che et al., 2019; Bilal
et al., 2018; Zhao et al., 2024; Huang et al., 2024). MCD12Q1 and MCD43A4 C6 were
already used by numerous studies (Giglio et al., 2018; Rodrigues et al., 2019; Zeng et
al., 2022; Wang et al., 2018). The multi-year aggregation of these products (Table 2)
further reduces the impact of the slight difference between these two versions.
Therefore, we think that the version difference will not make a significant impact on
MLA mapping. Following the comment, We have added these explanations to Section
2.2.1.
*We used MCD43A1 C6.1 and MCD12Q1 and MCD43A4 C6 for MLA mapping as*
*these data were available on GEE when this study was conducted. Only minor*

*calibration changes and polarization correction were adopted in the upgrading*
*from Collection 6 to 6.1, while the MCD12Q1 and MCD43A4 algorithms remain*
*the                                                                                   same*
*(https://landweb.modaps.eosdis.nasa.gov/data/userguide/MODIS_Land_C61_Ch*
*anges.pdf). In addition, the multi-year aggregation of these products (Table 2)*
*further mitigates the version impact.*

We agree with the referee that MODIS BRDF (MCD43A1) and surface reflectance
products (MCD43A4) used for MLA mapping (section 2.3.2) may be contaminated by
clouds, especially in tropical regions. MODIS BRDF is produced daily using multi-date,
cloud-cleared, atmospherically corrected input data measured over neighboring 16-day
periods (https://lpdaac.usgs.gov/products/mcd43a1v061/). When there is not enough
observation to derive BRDF robustly because of the cloud contamination, a backup
algorithm is employed which uses prior BRDF shapes and adjusts them with limited
observations. This study used all observations including low-quality backup BRDF
inversions. This practice has been adopted in global clumping index mapping with
BRDF products and a corresponding quality indicator has been provided (Wei et al.,
2019). Because we utilized the multi-year aggregation (10%, 33%, 50%, 67%, 90%
quantiles, and standard deviation, Table 2) of BRDF and surface reflectance in the MLA
mapping, the influence induced by low-quality inversions can be partly mitigated
(Sulla-Menashe et al., 2019). In response to the comment, we have added these
explanations to section 2.3.2.

*This study used all MODIS BRDF and spectral reflectance data including low-*
*quality ones that may be contaminated by clouds. The multi-year aggregation*
*(Table 2) can partly mitigate the influence induced by low-quality observations*
*(Sulla-Menashe et al., 2019).*

In addition, we have added a quality layer regarding the proportion of high-quality
BRDF inversions (see reply to comment #4 below).

4). As the first global MLA product, it would be valuable to include an uncertainty
assessment layer. This might account for the uncertainties stemming from (1) the
upscaling approach, (2) the machine learning model, and (3) data inputs. Presenting an
explicit uncertainty layer would markedly improve the credibility and potential
applications of this novel dataset.

We thank the referee for the recognition and excellent suggestion! We have
reconsidered the uncertainty sources of MLA mapping, including the upscaling
approach, data inputs, and machine learning model. The upscaling approach mainly
influences the uncertainty of training samples which is difficult to quantify for each
pixel. The rigorous sample screening after the upscaling process ensures the sample
representativeness (section 2.3.1) and reduces the uncertainty raised by the upscaling
process. The random forest algorithm is also robust to the remained sample uncertainty
as mentioned above.
Regarding model inputs, BRDF and BRDF-adjusted surface reflectance products are
important for MLA mapping (Fig. 6), and the same qualitative quality layer indicating
whether full BRDF inversions are provided for these products. This study used all
observations including low-quality backup BRDF inversions as stated above. Therefore,
we have added a quantitative quality layer to represent the proportion of high-quality
BRDF inversions for each pixel.
In terms of the prediction model, the machine learning model is typically regarded as a
black box, and evaluating the uncertainty for the random forest algorithm is difficult
under the current technological background. The random forest algorithm is accurate
enough for the predictions fall into the feature space ranges of training samples. For the
predictions out of the range of sample features, extrapolation is necessary and the
uncertainty is higher. Therefore, the prediction model quality was expressed
qualitatively for each pixel considering whether the MLA is predicted by extrapolating
beyond the range of the training samples.
Fig. R4 shows the quality layers regarding inputs and prediction model. The global
mean proportion of high-quality BRDF inputs is 68.03%. Northern South America and
Central Africa have a low proportion of high-quality inputs (20%) due to cloud
contamination (Fig. R4 (a)). Considering the large number of observations for each
pixel (7904 from 2001 to 2022), this percentage (20%) of high-quality observations is
sufficient to map MLA. In addition, 80.39% of the global MLA map was derived within
the feature ranges of training samples, and the rest 19.61% were mainly located in high-
latitude regions and Africa. For the latter areas, the MLA map was predicted with
extrapolation and caution should be taken when using the map (Fig. R4 (b)).

[Figure]

Fig. R4 Global distributions of quality indicators. (a) and (b) denote the proportion of high-quality BRDF inversions, and whether the prediction is within the ranges of training samples, respectively.

In response to the comment, we have added the contents regarding quality layers to Sections 2.3.2 and 3.3. In addition, the data products released on Zenodo have been updated (https://doi.org/10.5281/zenodo.12739662).

*Section 2.3.2*

*Two quality layers were added to represent the quality of input data and the prediction model. The input data quality was denoted by the proportion of high-quality BRDF inversions for each pixel. The prediction model quality was represented qualitatively for each pixel considering whether the MLA was predicted by extrapolating beyond the range of the training samples. The random*

*forest model is typically regarded as a black-box and its uncertainty is difficult to*
*quantify in the present study.*
*Section 3.3*
*Fig. 12 demonstrates the global distributions of the MLA quality indicators. The*
*global mean proportion of high-quality BRDF inputs is 68.03%. Northern South*
*America and Central Africa have a low proportion of high-quality inputs (20%)*
*because of cloud contamination (Fig. 12 (a)). Considering the large number of*
*observations for each pixel (7904 from 2001 to 2022), this percentage (20%) of*
*high-quality observations is sufficient to map MLA. In addition, 80.39% of the*
*global MLA map was derived within the feature ranges of training samples, and*
*the rest 19.61% were mainly located in high-latitude regions and Africa. For the*
*latter areas, the MLA map was predicted with extrapolation and caution should*
*be taken when using the map (Fig. 12 (b)).*

**Anonymous referee #3**

Thanks to the authors for the meticulous revisions. My key concerns have been addressed in this revised manuscript. I have one new suggestion. Although the authors have conducted experiments to prove that the nonlinear relationship between LAI (Leaf

Area Index) and EVI (Enhanced Vegetation Index) has little impact on the results, I

suggest that the reasons for the nonlinearity of LAI-EVI, especially the saturation phenomenon, should be elaborated in the text. In addition, why not use other vegetation indices such as NDVI (Normalized Difference Vegetation Index)? Since many papers on the error propagation of vegetation indices, the evaluation of saturation phenomena, and the relationships between vegetation indices and LAI and LCC (Leaf Chlorophyll

Content) have been published recently, it is recommended that the author explain why

EVI was chosen by citing such papers. Meanwhile, I suggest that the author incorporate more of the responses to the reviewers into the main body of the paper.

We thank the referee for the recognition and suggestion. The slight nonlinearity between LAI and EVI2 is induced by the saturation effect at medium and high LAI

conditions where the reflectance in near-infrared and red wavelength is stable (Gao et al., 2023).

In this study, EVI2 was used instead of other vegetation indices. Unlike NDVI, EVI2

is highly resistant to the saturation effect (Gao et al., 2023) and also shows a near-linear correlation with LAI (Dong et al., 2019; Alexandridis et al., 2019).

Following the suggestion, we have added these explanations to Section 2.3.1.

*Therefore, the 500 m MLA was computed as the weighted average of the enhanced*

*vegetation index (EVI2) assuming a linear relationship between LAI and EVI2*

*(Dong et al., 2019; Alexandridis et al., 2019). Although previous studies have*

*reported that vegetation index may be nonlinearly correlated to LAI because of*

*the saturation effect at medium and high LAI conditions, EVI2 is highly resistant*

*to the saturation effect (Gao et al., 2023). The errors caused by this slight*

*nonlinearity were further analyzed in Section 4.4.*

In addition, we have incorporated more of the responses to the reviewers into the main body of the paper in the revised version. Specifically, the responses regarding the spatial distribution and representativeness of samples (Section 2.3.1), the importance of biome map in MLA prediction (Section 4.2), the introduction of RS-based vegetation structure parameters as predictive variables (Section 4.4), and the choice of distance threshold (Section 2.3.1) have been further integrated.

**Reference**

Alexandridis, T. K., Ovakoglou, G., and Clevers, J. G. P. W.: Relationship between MODIS EVI and LAI across time and space, Geocarto International, 35, 1385-1399, 10.1080/10106049.2019.1573928, 2019.

Bilal, M., Nazeer, M., Qiu, Z., Ding, X., and Wei, J.: Global Validation of MODIS C6 and C6.1 Merged Aerosol Products over Diverse Vegetated Surfaces, Remote Sensing, 10, 10.3390/rs10030475, 2018.

Che, H., Yang, L., Liu, C., Xia, X., Wang, Y., Wang, H., Wang, H., Lu, X., and Zhang, X.: Long-term validation of MODIS C6 and C6.1 Dark Target aerosol products over China using CARSNET and AERONET, Chemosphere, 236, 124268, 10.1016/j.chemosphere.2019.06.238, 2019.

Chianucci, F., Pisek, J., Raabe, K., Marchino, L., Ferrara, C., and Corona, P.: A dataset of leaf inclination angles for temperate and boreal broadleaf woody species, Annals of Forest Science, 75, 50-50, 10.1007/s13595-018-0730-x, 2018.

de Wit, C. T.: Photosynthesis of leaf canopies, Pudoc, 1965.

Dong, T., Liu, J., Shang, J., Qian, B., Ma, B., Kovacs, J. M., Walters, D., Jiao, X., Geng, X., and Shi, Y.: Assessment of red-edge vegetation indices for crop leaf area index estimation, Remote Sens. Environ., 222, 133-143, 10.1016/j.rse.2018.12.032, 2019.

Dubayah, R., Blair, J. B., Goetz, S., Fatoyinbo, L., Hansen, M., Healey, S., Hofton, M., Hurtt, G., Kellner, J., Luthcke, S., Armston, J., Tang, H., Duncanson, L., Hancock, S., Jantz, P., Marselis, S., Patterson, P. L., Qi, W., and Silva, C.: The Global Ecosystem Dynamics Investigation: High-resolution laser ranging of the Earth's forests and topography, Science of Remote Sensing, 1, 10.1016/j.srs.2020.100002, 2020.

Fang, H., Baret, F., Plummer, S., and Schaepman-Strub, G.: An Overview of Global Leaf Area Index (LAI): Methods, Products, Validation, and Applications, Rev. Geophys., 57, 739-799, 10.1029/2018rg000608, 2019.

Gao, S., Zhong, R., Yan, K., Ma, X., Chen, X., Pu, J., Gao, S., Qi, J., Yin, G., and Myneni, R. B.: Evaluating the saturation effect of vegetation indices in forests using 3D radiative transfer simulations and satellite observations, Remote Sens. Environ., 295, 10.1016/j.rse.2023.113665, 2023.

Giglio, L., Boschetti, L., Roy, D. P., Humber, M. L., and Justice, C. O.: The Collection 6 MODIS burned area mapping algorithm and product, Remote Sens. Environ., 217, 72-85, 2018.

Huang, G., Su, X., Wang, L., Wang, Y., Cao, M., Wang, L., Ma, X., Zhao, Y., and Yang, L.: Evaluation and analysis of long-term MODIS MAIAC aerosol products in China, Sci. Total Environ., 948, 174983, 10.1016/j.scitotenv.2024.174983, 2024.

Moreno-Martínez, Á., Camps-Valls, G., Kattge, J., Robinson, N., Reichstein, M., van Bodegom, P., Kramer, K., Cornelissen, J. H. C., Reich, P., Bahn, M., Niinemets, Ü., Peñuelas, J., Craine, J. M., Cerabolini, B. E. L., Minden, V., Laughlin, D. C., Sack, L., Allred, B., Baraloto, C., Byun, C., Soudzilovskaia, N. A., and Running, S. W.: A methodology to derive global maps of leaf traits using remote sensing and climate data, Remote Sens. Environ., 218, 69-88, 10.1016/j.rse.2018.09.006, 2018.

Pisek, J., Ryu, Y., and Alikas, K.: Estimating leaf inclination and G-function from leveled digital camera photography in broadleaf canopies, Trees, 25, 919-924, 10.1007/s00468-011-0566-6, 2011.

Rodrigues, J. A., Libonati, R., Pereira, A. A., Nogueira, J. M., Santos, F. L., Peres, L. F., Santa Rosa, A., Schroeder, W., Pereira, J. M., and Giglio, L.: How well do global burned area products represent fire patterns in the Brazilian Savannas biome? An accuracy assessment of the MCD64 collections, International Journal of Applied Earth Observation and Geoinformation, 78, 318-331, 2019.

Ryu, Y., Sonnentag, O., Nilson, T., Vargas, R., Kobayashi, H., Wenk, R., and Baldocchi, D. D.: How to
quantify tree leaf area index in an open savanna ecosystem: A multi-instrument and multi-model
approach, Agricultural and Forest Meteorology, 150, 63-76, 10.1016/j.agrformet.2009.08.007, 2010.

Siegmund, A. and Menz, G.: Fernes nah gebracht–Satelliten-und Luftbildeinsatz zur Analyse von
Umweltveränderungen im Geographieunterricht, Geographie und Schule, 154, 2-10, 2005.

Sulla-Menashe, D., Gray, J. M., Abercrombie, S. P., and Friedl, M. A.: Hierarchical mapping of annual
global land cover 2001 to present: The MODIS Collection 6 Land Cover product, Remote Sens. Environ.,
222, 183-194, 10.1016/j.rse.2018.12.013, 2019.

Svetnik, V., Liaw, A., Tong, C., Culberson, J. C., Sheridan, R. P., and Feuston, B. P.: Random forest: a
classification and regression tool for compound classification and QSAR modeling, Journal of chemical
information and computer sciences, 43, 1947-1958, 2003.

Tang, H., Ganguly, S., Zhang, G., Hofton, M. A., Nelson, R. F., and Dubayah, R.: Characterizing leaf
area index (LAI) and vertical foliage profile (VFP) over the United States, Biogeosciences, 13, 239-252,
10.5194/bg-13-239-2016, 2016.

Wang, Z., Schaaf, C. B., Sun, Q., Shuai, Y., and Román, M. O.: Capturing rapid land surface dynamics
with Collection V006 MODIS BRDF/NBAR/Albedo (MCD43) products, Remote Sens. Environ., 207,
50-64, 2018.

Wei, S., Fang, H., Schaaf, C. B., He, L., and Chen, J. M.: Global 500 m clumping index product derived
from MODIS BRDF data (2001–2017), Remote Sens. Environ., 232, 111296,
https://doi.org/10.1016/j.rse.2019.111296, 2019.

Yan, G., Jiang, H., Luo, J., Mu, X., Li, F., Qi, J., Hu, R., Xie, D., and Zhou, G.: Quantitative Evaluation
of Leaf Inclination Angle Distribution on Leaf Area Index Retrieval of Coniferous Canopies, Journal of
Remote Sensing, 2021, 1-15, 10.34133/2021/2708904, 2021.

Zeng, Y., Hao, D., Huete, A., Dechant, B., Berry, J., Chen, J. M., Joiner, J., Frankenberg, C., Bond-
Lamberty, B., and Ryu, Y.: Optical vegetation indices for monitoring terrestrial ecosystems globally,
Nature Reviews Earth & Environment, 3, 477-493, 2022.

Zhao, R., Yu, W., Deng, X., Huang, Y., Yang, W., and Zhou, W.: Analysis of Land Surface Performance
Differences and Uncertainty in Multiple Versions of MODIS LST Products, Remote Sensing, 16,
10.3390/rs16224255, 2024.

Zou, X., Mõttus, M., Tammeorg, P., Torres, C. L., Takala, T., Pisek, J., Mäkelä, P., Stoddard, F. L., and
Pellikka, P.: Photographic measurement of leaf angles in field crops, Agricultural and Forest Meteorology,
184, 137-146, 10.1016/j.agrformet.2013.09.010, 2014.

---

## Author Response (AR4)

**Topic editor**

Public justification (visible to the public if the article is accepted and published):

An additional reviewer was invited to evaluate the revised version of the manuscript.

This new reviewer has provided a set of minor suggestions to further improve the clarity and quality of the paper. The authors are encouraged to carefully address these new comments and incorporate the necessary revisions to enhance the overall readability and scientific rigor of the manuscript.

We greatly appreciate the topic editor for his professional handling throughout the entire process. In this round of revision, we have carefully addressed these comments raised by the new reviewer. Some minor revisions have been made in the revised version and please see the itemized responses below. We hope that this version can reach the publishing level of ESSD.

**Referee #4**

This article presents a comprehensive study on generating the first global 500 m mean

LIA (Mean Leaf Area) product using field measurements and remote sensing data. It can improve understanding of global LIA distribution and enhance applications in radiative transfer modeling, remote sensing, and land surface modeling.

The manuscript is well-structured, with clear objectives, thorough data collection, and robust analytical methods. The results are presented in a detailed and comprehensible manner. It is noted that this is the third revised version. Based on the authors' previous responses, they have generally addressed the reviewers' comments. I only have a few additional comments for consideration:

We thank the referee for the recognition and detailed comments which further improved the manuscript. We fully understand the referee's comments and have provided detailed explanations and revisions below.

1. Validation of G(0). In line 442, it states, "Fig. 14 shows that most of the reference

G(0) values are greater than 0.50." However, it is not immediately clear from Fig. 14.

Indeed, the predicted G(0) values appear to be predominantly greater than 0.5. In

Section 4.1, the authors suggest that this may be attributed to the CI values, which seems reasonable. However, if I understand correctly, the G(0) value is also influenced by the FVC (Fractional Vegetation Cover) value (BTW, I recommend that the authors define the concept of FVC in the manuscript). The effects of both CI and FVC

contribute to the uncertainty in the validation results. Given that a significant finding of this study is that G(0) exceeds 0.5 in most cases, I suggest improvements could be made to this assessment.

Furthermore, the phrase "while the spherical distribution would underestimate the interception of radiation and rainfall (Figs. 9 and 11)" requires clarification, as the logic is not easily discernible from Figs. 9 and 11.

We thank the referee for the comments. In Fig. 14, it is not obvious that most of the reference G(0) values are greater than 0.50 because we show the validation compassion with an error bar for each site to better show the difference between different sites. The data statistics demonstrate that 72% of data points have G(0) values greater than 0.50.

In response to the comment, we have added the alternative form of Fig. 14 (Fig. R1) to the supplement and rephrased the original sentences in line 440.

*In addition, most (72%) of the reference G(0) values are greater than 0.50 (Fig. S8),……*

[Figure]

Fig. R1 Comparisons of G(0) derived from mean leaf inclination angle and high-resolution reference data for different plant functional types (see Fig. 2 for the acronyms).

In addition, the definition of FVC has been added to line 147. In this study, CI and FVC were used to derive high-resolution reference G(0) but not to predict the G(0) product. The CI angular effect may have caused the possible bias in the reference G(0) because it was ignored in calculating the reference G(0) (Eqs. (6) and (7)). In addition, the high-resolution FVC and LAI products may be influenced by woody materials that were included in the field measurements. We have discussed these points and added related sentences (line 396).

*The woody materials may introduce biases into the reference G(0) as they were not separated in the high-resolution FVC and LAI products. The mixture of woody materials and leaves may have caused the underestimation of the reference G(0) because trunks usually have higher inclination angles (Liu et al., 2019).*

The spherical distribution may underestimate the interception of radiation and rainfall because it overestimates LIA and underestimates G(0) for most conditions as shown in Figs. 9 and 11. We have rephrased the original sentences (line 440).

*in this case, the spherical distribution would underestimate the radiation and rainfall interception because of the overestimated LIA and underestimated G(0) for most conditions (Figs. 9 and 11) (Stadt and Lieffers, 2000)*

2. Lines 415-417: Please check the statements made here. They may not be suitable for all cases.

We agree with the reviewer that this statement is not rigorous. We have rephrased this sentence in a more rigorous manner (line 413).

*Higher MLA generally means lower canopy interception and higher transmission for high solar altitude and more soil background can be detected in the nadir direction (Liu et al., 2012). This results in lower (higher) canopy NIR (red) reflectance because of the generally lower (higher) NIR (red) soil reflectance than that of the leaf components (Siegmund and Menz, 2005) and negative correlations between MLA and NIR reflectance and NDVI (Liu et al., 2012).*

3. Lines 23-24: The phrase "is opposite" is not clear. And the unit for the RMSE value is missing.

The global G(0) distribution is out of phase with that of the MLA and we have rephrased that in the text. The G(0) is unitless and thus its RMSE has no unit.

4. Both Sections 2.1.1 and 2.1.2 reference Figure 1. However, there is a lack of necessary differentiation and explanation regarding this figure within the text and the figure itself.

Figure 1 includes all LIA field measurements with location information from TRY and literature and thus was refer in Sections 2.1.1 and 2.1.2. We have added the related explanation to this figure.

*Figure 1. The locations of global leaf inclination angle measurements collected*
*from TRY and the literature.*

5. Lines 97-99: The description stating, "The LIA measurements in published literature
were … from the literature (Fig. 1)" is overly simplistic. More detailed information
should be provided to guide readers on how to access these measurements, thereby
enhancing the credibility and reproducibility of the study.

We thank the referee for this point. We have added the necessary descriptions to line
96.
*To fully utilize distributed and considerable LIA measurement data in the*
*published literature, several keyword searches (leaf angle, leaf inclination angle,*
*and leaf tilt angle) were performed in the Web of Science, Google Scholar, Google,*
*and Chinese documentary databases.*

6. Lines 142-143: The MCD15A2H product is stated to be available only from July
2002. Please clarify how data prior to July 2002 is obtained for this analysis.

Indeed, the MCD15A2H product is only available from July 2002. This study used the
product since 2002. This product was only used for G(0) upscaling validation but not
for MLA mapping. In addition, the multi-year average from 2002-2022 also reduces the
impact of the lack of one year.

7. Section 2.2.2 - High-resolution Reference Data: It would be beneficial to move some
of the descriptions from Section 2.4 to this section for a better logic, as it may not be
easy for readers to follow otherwise.

We thank the referee for this comment. Following the comment, we have moved the
high-resolution reference G(0) from Section 2.4 to Section 2.2.2 to enhance structural
coherence.

8. Equation (3): This equation may require a citation to a relevant reference.

We have added a relevant citation for Equation (3).
*The G(θ) value in the nadir direction (θ=0°) was calculated using an analytical*
*formula (Leblanc and Fournier, 2017).*